# Whole-soil warming leads to substantial soil carbon emission in an alpine grassland

Ying Chen [1], Wenkuan Qin[1], Qiufang Zhang[1], Xudong Wang [1], Jiguang Feng [1], Mengguang Han [1], Yanhui Hou[1], Hongyang Zhao[1], Zhenhua Zhang [2], Jin-Sheng He [1,3], Margaret S. Torn [4,5] & Biao Zhu [1] ✉

The sensitivity of soil organic carbon (SOC) decomposition in seasonally frozen soils, such as alpine ecosystems, to climate warming is a major uncertainty in global carbon cycling. Here we measure soil $CO_2$ emission during four years (2018–2021) from the whole-soil warming experiment (4 °C for the top 1 m) in an alpine grassland ecosystem. We find that whole-soil warming stimulates total and SOC-derived $CO_2$ efflux by 26% and 37%, respectively, but has a minor effect on root-derived $CO_2$ efflux. Moreover, experimental warming only promotes total soil $CO_2$ efflux by 7-8% on average in the meta-analysis across all grasslands or alpine grasslands globally (none of these experiments were whole-soil warming). We show that whole-soil warming has a much stronger effect on soil carbon emission in the alpine grassland ecosystem than what was reported in previous warming experiments, most of which only heat surface soils.

Soil is the largest carbon pool in terrestrial ecosystems (1500–2400 Pg, over two times the size of the atmospheric carbon pool and terrestrial plant carbon pool), playing a key role in regulating global carbon cycling[1,2]. Even a small increase in the carbon efflux derived from soil organic carbon (SOC) decomposition could have a large effect on atmospheric carbon dioxide ($CO_2$) concentrations[3]. There is considerable concern that climate warming will accelerate the decomposition of soil carbon and increase the soil $CO_2$ efflux to the atmosphere, thereby creating a positive carbon-climate feedback[4–6]. However, there is large uncertainty in the amount and rate of SOC response to warming, as the balance of C inputs from plant residues and C outputs via microbial decomposition to increasing temperature is not clear in the whole-soil profile.

According to the soil temperature predictions from IPCC models, both surface and deep soils will be warmed by approximately 4 °C at the same rate, following the air warming trends under Representative Concentration Pathway (RCP) 8.5 by the end of the 21st century[7]. Most

of the previous in situ warming experiments could only warm the surface soil (0–20 cm), but not the subsoils or deep soils (>20 cm), which contain about 50% of the total carbon stocks[8,9]. The lack of data on deep soils in response to in situ field warming and thus the lack of whole-soil carbon response to warming can lead to large uncertainty in model predictions of global carbon cycling. The whole-soil warming experiment in a temperate forest found that two years of warming increased $CO_2$ release from all soil depths, mainly from free-light SOC (decadal-aged carbon), more than field warming experiments and models that only warmed surface soils[6]. The 5-year results from the same experiment showed that the increase in soil $CO_2$ efflux was associated with a loss of deep soil carbon[10]. The amount of soil carbon loss due to warming is related to not only increased soil carbon emissions by microbes, but also warming-induced altered plant inputs[11,12]. The 2-year whole-soil warming in a tropical forest significantly increased soil carbon emissions, with the increase mainly originating from heterotrophic respiration rather than autotrophic

[1]Institute of Ecology and Ministry of Education Key Laboratory for Earth Surface Processes, College of Urban and Environmental Sciences, Peking University, Beijing 100871, China. [2]Qinghai Haibei National Field Research Station of Alpine Grassland Ecosystem and Key Laboratory of Adaptation and Evolution of Plateau Biota, Northwest Institute of Plateau Biology, Chinese Academy of Sciences, Xining 810008, China. [3]State Key Laboratory of Herbage Improvement and Grassland Agro-ecosystems and College of Pastoral Agricultural Science and Technology, Lanzhou University, Lanzhou 730000, China. [4]Climate and Ecosystem Sciences, Lawrence Berkeley National Laboratory, Berkeley, CA 94720, USA. [5]Energy and Resources Group, University of California, Berkeley, Berkeley, CA 94720, USA. ✉e-mail: biaozhu@pku.edu.cn

respiration[13]. Moreover, no physiological adaptation of microbes was found, and microbially mediated soil carbon release continued, resulting in a significant positive feedback to climate warming[13]. High-latitude and high-altitude ecosystems represent key ecosystems for the evaluation of warming effects[3,14,15] because they are warming quickly[16]. For example, the Tibetan Plateau is warming twice (0.3–0.4 °C per decade) as fast as the global average[17]. Importantly, deep soil layers will become active in C cycling in these cold environments, as these layers would be warmer and have more time to thaw each year. Therefore, understanding and quantifying the response of the whole soil to warming is needed in high-latitude and high-altitude ecosystems.

Here we present results from the whole-soil warming experiment in an alpine grassland ecosystem on the Tibetan Plateau (Supplementary Fig. 1). The experiment explored the response of the whole-soil profile (0–100 cm; mainly soil carbon emissions, but also plant and soil properties) to 4 °C warming over 4 years (2018–2021). This field warming experiment consists of four pairs of circular control and warming plots. In warming plots, twenty 1 m deep heating cables were installed vertically around 3 m diameter plots, and 1 m and 2 m diameter heating cables were placed horizontally on the soil surface at 5 cm depth. With a whole-soil warming target of 4 °C, the subsoils (10–100 cm) were essentially able to achieve 4 °C of warming and the surface soil (0–10 cm) temperature increased by 3.1 °C (Fig. 1a and Supplementary Fig. 2). Moreover, a meta-analysis was made to compare the response of soil $CO_2$ efflux to warming from our whole-soil warming experiment in an alpine grassland with the general response pattern in global grasslands derived from traditional surface-soil warming experiments. In this

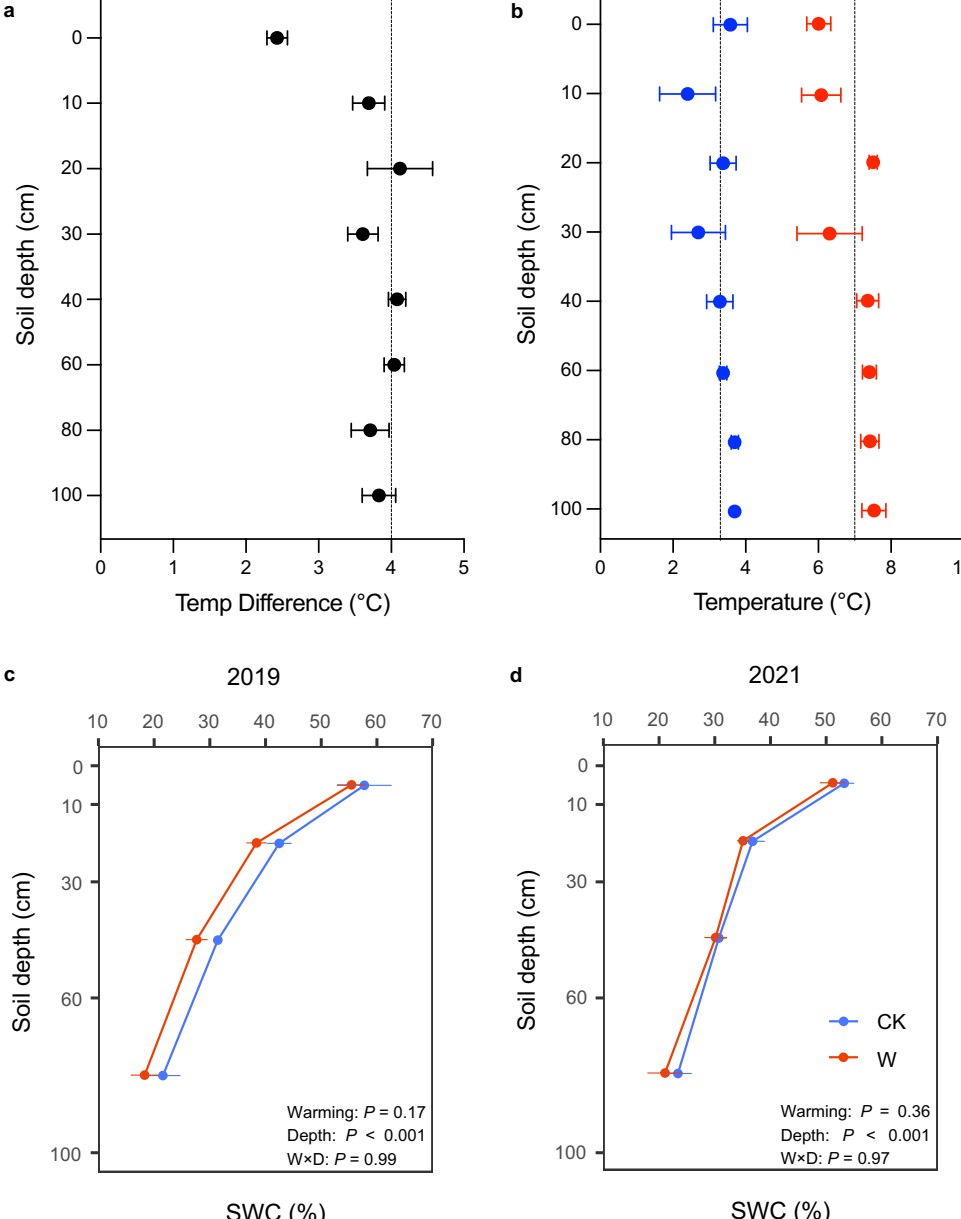

**Fig. 1 | Soil temperature and water content across the soil profile (0–100 cm) over four years (measured continuously from June 2018 to September 2021).** **a** Mean soil temperature difference (warming−control). **b** Mean soil temperature. The vertical lines show the mean soil temperature over the whole soil profile, which is 3.3 °C for control treatment and 7.0 °C for warming treatment. Soil water content (SWC) was measured on samples in August 2019 (**c**) and August 2021 (**d**). We use the linear mixed-effect models to estimate the effects of warming on soil water content across soil profile (0–100 cm). Different colors mean different treatments (blue indicates control, and red indicates warming). Bar values are the mean ± standard error (*n* = 4, biologically independent samples). Source data are provided as a Source Data file.

work, we show that whole-soil warming increased total soil $CO_2$ efflux (mainly from SOC) from the alpine grassland much higher than the average effect size of none-whole-soil warming in the meta-analysis across global grasslands, indicating that whole-soil warming could induce more soil carbon emission in the future.

## Results

### Whole-soil warming effects on microclimates, plant and soil properties

In this field study, we aimed to warm the whole-soil profile (0–100 cm) by 4 °C (Supplementary Fig. 2). The soils from 10 to 100 cm were warmed by close to 4 °C (Fig. 1a). Overall for the 4 years (2018–2021), the average temperature was 3.3 °C for control treatments and 7.0 °C for warming treatments with a similar average temperature for different soil depths (Fig. 1b). Although we buried 2 concentric rings of heating cables in the surface soil layer (5 cm), we did not achieve the temperature increase target over four years (from June 2018 to September 2021, only 2.4 and 3.7 °C for 5 and 10 cm depth, respectively) (Fig. 1a). Soil moisture showed little response to whole-soil warming, either in volumetric (continuously measured, variable among plots and layers, Supplementary Fig. 3) or gravimetric (measured during soil sampling, Fig. 1c, d) water content.

Plants are the ultimate source of soil carbon inputs, therefore we also determined the responses of plant properties (including ANPP [aboveground net primary productivity], AGB [aboveground biomass], BNPP [belowground net primary productivity], and BGB [belowground biomass]) to the whole-soil warming over four years (2018-2021). Based on the 4-year results, we found that whole-soil warming had a minor effect on plant biomass (Supplementary Fig. 4). ANPP, AGB, BNPP, and BGB were mostly not significantly affected by whole-soil warming during 2018-2021, except for ANPP and AGB in 2018 and AGB in 2020 (Supplementary Figs. 4, 5). We also tested the responses of soil nutrient, microbial, and carbon properties ($NH_4^+$-N [ammonium nitrogen], $NO_3^-$-N [nitrate nitrogen], EOC [extractable organic carbon], ETN [extractable total nitrogen], MBC [microbial biomass carbon], MBN [microbial biomass nitrogen], BG [BG, β-1, 4-glucosidase], NAG [β-1, 4-N-acetyl-glucosaminidase], AP [acid phosphatase], and CUE [carbon use efficiency]), and found that all variables significantly decreased with increasing soil depth (sampled in 2019 and 2021, Supplementary Figs. 6, 7). But, warming did not significantly alter SOC content, either from the whole-soil profile or from a specific soil depth (0–10, 10–30, 30–60, 60–100 cm) (Supplementary Fig. 7a). Similarly, the effects of whole-soil warming on most variables were also not significant after 14 months (from June 2018 to August 2019) and 38 months (from June 2018 to August 2021) of warming treatment (Supplementary Figs. 6, 7).

### Whole-soil warming effects on soil $CO_2$ efflux

In the pre-treatment period, soil respiration (Rs, total), heterotrophic respiration (Rh, SOC-derived), and autotrophic respiration (Ra, root-derived) did not differ significantly between the warming and control plots (Figs. 2 and 3a). For the overall results of 4 years (31 sampling dates), we found significant effects of warming on total and SOC-derived soil $CO_2$ efflux ($P = 0.01$ and $P < 0.001$), while there was no significant effect of warming on autotrophic respiration (Fig. 2). There were significant temporal fluctuations in all types of soil $CO_2$ efflux (total, SOC-derived, and root-derived, $P < 0.001$). The interaction between warming and time significantly affected root-derived $CO_2$ efflux ($P = 0.04$), while it had no effects on total and SOC-derived $CO_2$ efflux (Fig. 2). Overall, whole-soil warming significantly elevated total and SOC-derived soil $CO_2$ efflux over time (but not root-derived $CO_2$ efflux, Fig. 2; Bonferroni test, $P < 0.001$). In each year (growing-season) after the warming treatment commenced, either 2018, 2019, 2020, or 2021, we found that warming elevated total and SOC-derived $CO_2$ efflux ($P < 0.10$ in 2018, $P < 0.05$

in 2019-2021, Supplementary Fig. 8). However, root-derived $CO_2$ efflux did not respond to warming in any year (Supplementary Fig. 8). Four years of whole-soil warming significantly increased total soil $CO_2$ emissions by 26%, from 3.99 μmol m$^{-2}$ s$^{-1}$ in control plots to 5.05 μmol m$^{-2}$ s$^{-1}$ in warming plots ($P = 0.01$, Fig. 3b). Moreover, increased soil $CO_2$ efflux was mainly from soil heterotrophic respiration (SOC-derived, microbial decomposition of soil organic carbon, Fig. 3b). The SOC-derived respiration increased from 2.40 μmol m$^{-2}$ s$^{-1}$ in control plots to 3.30 μmol m$^{-2}$ s$^{-1}$ in warming plots (a 37% increase of 0.90 μmol m$^{-2}$ s$^{-1}$, $P = 0.02$), but the root-derived respiration was not altered significantly (1.60 and 1.79 μmol m$^{-2}$ s$^{-1}$ in control and warming plots, respectively, Fig. 3b). Moreover, the response of soil $CO_2$ efflux to whole-soil warming after correction (to reduce spatial heterogeneity) was similar to that without correction (Supplementary Fig. 9).

However, the warming-induced change in total soil respiration, SOC-derived respiration or root-derived respiration did not vary significantly among the four years ($P > 0.05$, Fig. 4). We also calculated the contribution of SOC-derived and root-derived components to total soil $CO_2$ efflux during the four years. Despite temporal fluctuations ($P < 0.001$), there was no significant warming effect nor the interaction between warming treatment and time on the root or soil (microbial) contributions to total $CO_2$ efflux (Supplementary Fig. 10). The average results of 4 years showed no significant effect of warming treatment on the relative proportions of soil $CO_2$ emissions from either SOC-derived or root-derived components (Supplementary Fig. 11).

### Synthesis of warming effects on soil $CO_2$ efflux across grasslands

In the meta-analysis across global grasslands, the effects of experimental warming (none were whole-soil warming) on different types of soil $CO_2$ efflux (Rs, total; Rh, SOC-derived; Ra, root-derived) were diverse (Fig. 5a and Supplementary Fig. 12). Warming significantly increased Rs by 7% ($n = 234$, 95% CI: 1%–13%, total data), but the responses of Rs ($n = 46$, paired data), Rh and Ra ($n = 46$, for both total and paired data) to warming were not significant (Supplementary Fig. 12). In addition, different types of soil $CO_2$ efflux (Rs, Rh, and Ra) from alpine grasslands (none were whole-soil warming) did not respond significantly to experimental warming (Fig. 5b). According to the results of model-averaged relative importance of predictors, the duration of warming experiments could regulate the response of Rs (total data) to warming across grasslands globally (Supplementary Fig. 13). Additionally, the effect size of Rs (total data, not the paired data) had a significant positive relationship with the duration of warming (Supplementary Fig. 14a).

## Discussion

Averaged over the four years, whole-soil warming markedly increased total and SOC-derived $CO_2$ efflux by 26% and 37%, respectively, but had a weaker effect on root-derived $CO_2$ efflux (12%, Figs. 2, 3). Similar to our study in the alpine grassland, whole-soil warming experiments have also been conducted in temperate and tropical forests to study soil carbon emissions. In a temperate forest, Pries et al. found[6] that whole-soil warming caused larger soil respiration by 34%, and after 4.5 years of warming, there was still a 30% increase in soil $CO_2$ efflux[10] (Supplementary Table 1). A whole-soil warming experiment in a tropical forest found that 2 years of warming stimulated soil respiration by 55%[13] (Supplementary Table 1). Prior to these whole-soil warming experiments, most warming experiments could only warm the surface soils (0–20 cm), while the effects of previous warming on soil respiration were diverse. In a warming experiment conducted in an alpine meadow using infrared heaters to increase surface soil temperature (by 2.5 °C), it was found that warming increased soil respiration by about 14%[18], while a montane meadow experiment using the same technology found that soil respiration decreased by 8% due to 1.6 °C warming[19]. In addition, in two recent regional-scale meta-analysis

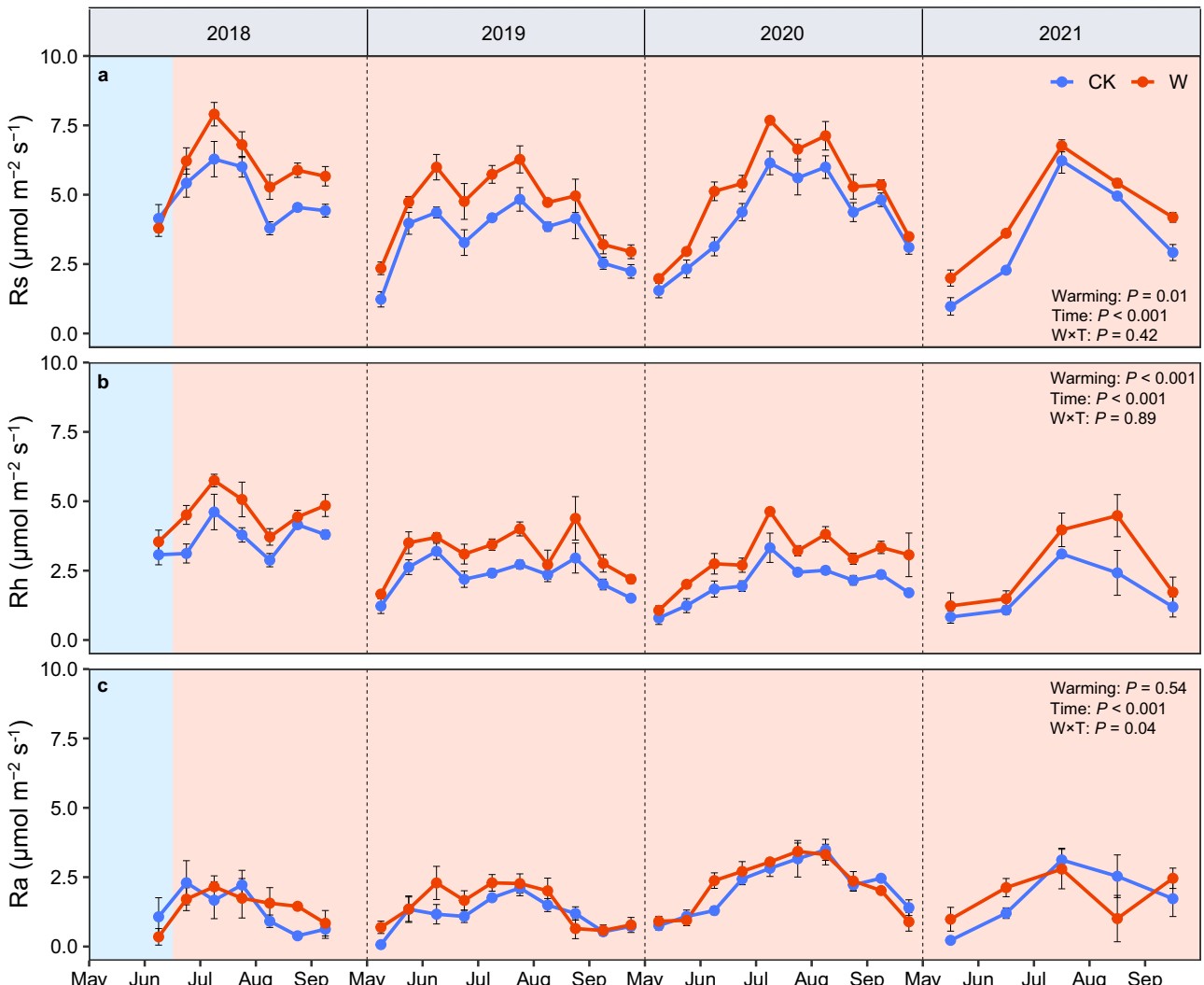

**Fig. 2 | Soil CO₂ efflux from control and warming treatments over four years (from June 2018 to September 2021). a** Rs (soil respiration, total). **b** Rh (soil heterotrophic respiration, SOC-derived). **c** Ra (soil autotrophic respiration, root-derived). The repeated measures ANOVA was used to test the effects of warming treatment and time on soil $CO_2$ efflux. In 2018-2020 growing seasons, measurements were made every two weeks; in 2021 growing season, measurements were made monthly. The light-blue shaded areas represent the pre-treatment period (before June 17, 2018) and the orange shaded areas represent four years of warming periods (from June 2018 to September 2021). Different colors mean different treatments (blue indicates control and red indicates warming). Bar values are the mean ± standard error ($n = 4$, biologically independent samples). Source data are provided as a Source Data file.

studies, warming promoted soil respiration by about 12% for alpine meadow ecosystems[15], while warming also elevated soil respiration by about 14% for alpine grassland ecosystems[20]. Similarly, experimental warming elevated soil respiration by 7% on average across all grasslands (8% but non-significant for alpine grasslands) globally in our meta-analysis (total data, Fig. 5). Compared to these surface-soil warming studies, our study showed that whole-soil warming accelerates the decomposition of soil organic carbon across the soil profile, thus leading to more soil carbon release from the alpine grassland than the more common warming experiments that only warmed the topsoils[6,10,13].

Three mechanisms have been proposed to explain the warming-caused increase in soil CO₂ efflux to the atmosphere in previous studies. First, warming may elevate soil substrate and nutrient availability, and also promote plant growth, leading to increased soil organic carbon input and thus accelerating soil respiration [without changing rate constants for CO₂ production or microbial efficiency][21,22]. Second, warming could increase the biomass and activity of soil microorganisms, and thereby accelerate soil organic matter decay, promoting soil

respiration[23,24]. Third, warming could promote the belowground allocation of plant carbon inputs, and elevate belowground root biomass and activity, leading to an increase in root respiration and thereby enhancing soil respiration[25,26]. Our study was able to distinguish among these. Four years of whole-soil warming in the alpine grassland revealed that warming led to increased soil respiration mainly due to increased SOC-derived CO₂ efflux rather than root-derived CO₂ efflux (Figs. 2, 3). However, neither heterotrophic nor autotrophic respiration responded to experimental warming in the meta-analysis of global grasslands or alpine grasslands that used surface-soil warming (Fig. 5). In our study, there are three possible explanations for the large soil CO₂ efflux under whole-soil warming. (i) More available substrates due to warming would increase the activity of soil microorganisms to break down more soil organic matter along the soil profile (plant biomass tended to increase, Supplementary Figs. 4, 5), leading to the large release of soil CO₂. (ii) An acceleration of physiological activity and turnover of soil microbes due to warming (although microbial biomass and enzyme activity did not change significantly, Supplementary Figs. 6, 7) would also lead to a higher release of CO₂[27]. (iii) Perhaps

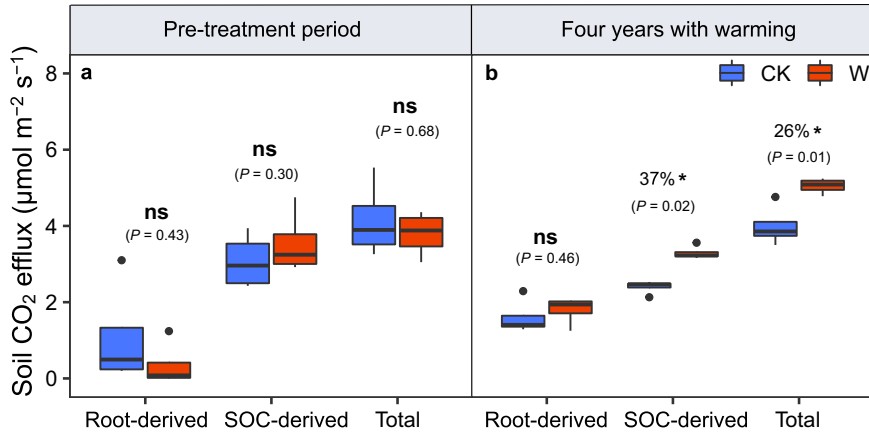

**Fig. 3 | The mean soil $CO_2$ efflux partitioned into SOC-derived (heterotrophic) respiration) and root-derived (autotrophic respiration) components for growing season measurements. a** The pre-treatment period (before June 17, 2018). **b** Four years of warming periods (from June 2018 to September 2021). Different colors mean different treatments (blue indicates control and red indicates warming). A paired *t*-test (two-sided) was used to determine the effects of warming treatments on mean soil $CO_2$ efflux. Statistical significance of differences between control and warming treatment is shown by asterisks († $P < 0.10$, * $P < 0.05$, ** $P < 0.01$, *** $P < 0.001$, $n = 4$, biologically independent samples) or as non-significant (ns). Boxes represent the interquartile range (IQR), and whiskers indicate the furthest point within $1.5 \times$ IQR above or below the IQR. Values beyond this range are plotted as individual points. The central line indicates the median. Source data are provided as a Source Data file.

more importantly, increased temperature could lengthen the thaw-season of soils (alpine grassland soils are frozen for most of the non-growing season), thus extending the growing or "decomposition" season that is suitable for microbial growth (Supplementary Fig. 2), increasing total losses of soil $CO_2$[20,28]. Notably, our results that whole-soil warming significantly elevated soil $CO_2$ efflux were obtained from only one whole-soil warming experiment in an alpine grassland which cannot represent global grassland ecosystems. The response of soil $CO_2$ efflux to whole-soil warming may vary depending on environmental factors, soil types, and plant and soil microbial communities, which requires verification by future coordinated distributed experiments with similar designs in other ecosystems and sites.

The warming-induced increase in soil $CO_2$ efflux did not attenuate over the four years of this whole-soil warming experiment. Substrate limitation and adaptation of soil microbial communities have been reported to occur after long-term warming[5,29]. In this study, we found no significant reduction in extractable or available carbon or nitrogen with warming (Supplementary Figs. 6, 7), as would be expected under nutrient limitation. Microbial CUE, which can reflect the change of microbial community and influence long-term soil carbon loss, was unaffected by whole-soil warming (Supplementary Figs. 6-7). In the meta-analysis, we also found that the effect of experimental warming on total soil $CO_2$ efflux was positively related to warming duration (Supplementary Figs. 13, 14, total data). However, the warming magnitude unexpectedly did not significantly change the response of total soil $CO_2$ efflux to warming, likely because the meta-analysis results were derived from different warming experiments (with different methods, durations, and climates), as similar results with previous studies[30,31]. Maybe the most important reason is that over 55% of the warming duration in this meta-analysis is less than or equal to 2 years. Therefore, the long-term patterns of soil $CO_2$ emissions and soil C stocks in response to warming need to be further investigated[12]. Notably, in addition to the duration of the experiment, the "phase" of the experiment[5] may also be important for interpreting the effects of climate change on ecosystem processes, since the experiment has just started and a drastic change in the environment, such as a whole-soil warming, will lead to certain adaptation effects. Together, these results suggest that whole-soil warming will likely continue to stimulate the degradation of soil carbon and cause more $CO_2$ emission in the alpine grassland, although longer-term measurements are required to test this.

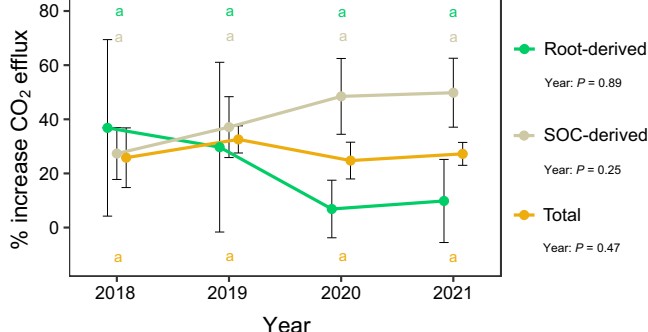

**Fig. 4 | The effect of warming on mean soil $CO_2$ efflux (in % increase) for growing season measurements over four years (from June 2018 to September 2021).** Different colors mean different sources (Total, SOC-derived, Root-derived). The repeated measures ANOVA was used to test the effect of year on mean soil $CO_2$ efflux. Bar values are the mean ± standard error ($n = 4$, biologically independent samples). The same lowercase letters of different colors indicate lack of significant differences among different years ($P > 0.05$) for each flux. Source data are provided as a Source Data file.

Alternatively, synchronous monitoring of soil temperature and soil carbon cycling from existing long-term experiments would provide a unique dataset to explore the effects of climate change on soil carbon dynamics.

Additionally, there are some points to be considered for future studies to reduce uncertainty in the climate-carbon feedback. First, we should focus on the release of $CO_2$ from all soil depths and their responses to whole-soil warming[10]. Second, many high latitude and high elevation experiments, including this study, measured soil respiration only during the growing season. More attention is needed for soil $CO_2$ emissions in the non-growing season, not just the growing season, because the former may be significant for the annual carbon budget[32,33]. Finally, we used a year-round constant temperature increase (4 °C) in this study, but there would be seasonal fluctuations in the temperature increase in different regions[34]. Therefore, we need to set up seasonal asymmetric warming experiments to study the response of ecosystems in the future.

More attention is needed on SOC-derived (i.e., heterotrophic) respiration rather than total or root-derived respiration because only

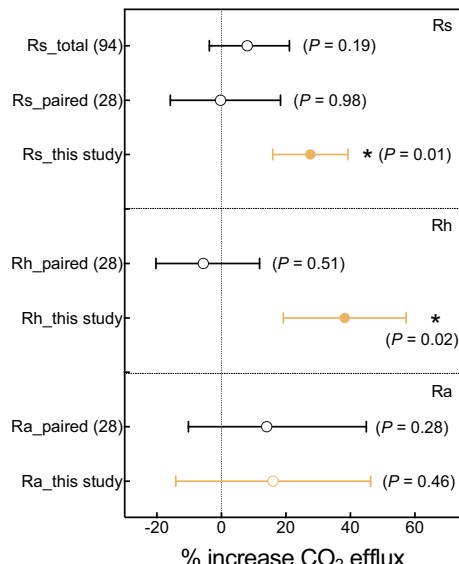

**Fig. 5 | The effect of warming on soil CO$_2$ efflux (and its components) from the meta-analysis of all grasslands or alpine grasslands globally as well as the whole-soil warming experiment in an alpine grassland.** The responses of soil respiration (Rs), heterotrophic respiration (Rh), and autotrophic respiration (Ra) to experimental warming (% increase CO$_2$ efflux) in all grasslands (a) or alpine grasslands (b) globally (all are surface-soil warming; meta-analysis, total and paired data) and this whole-soil warming in the alpine grassland. Black and yellow represent the results from the meta-analysis and our whole-soil warming experiment, respectively. Circles and error bars represent average parameter estimates and 95%

confidence intervals (CIs) in the linear mixed effects models (two-sided). The vertical dashed line represents the warming effect size = 0. If the 95% CI did not overlap zero, the effect of warming was statistically significant (denoted by *). Total data are from experiments that only measured total soil respiration (but did not separate it into heterotrophic and autotrophic components), while paired data are from experiments that separated total soil respiration into heterotrophic and autotrophic components. The sample size ($n$) for each variable is in parentheses. Source data are provided as a Source Data file.

SOC-derived CO$_2$ emissions directly lead to loss of soil carbon[3]. Our result is similar to that of Nottingham et al. [13]. With the finding that the cause of larger soil carbon loss by warming in a tropical forest was a marked increase in SOC-derived respiration rather than being related to root-derived sources, while Pries et al. and Soong et al. only focused[6,10] on total CO$_2$ efflux but not its sources. However, another study showed the opposite result: although OTC warming stimulated soil respiration, it did not change heterotrophic respiration but only increased autotrophic respiration, which was an important reason why soil organic carbon did not change after warming in alpine meadows[15]. Our results also showed that the warming effect on heterotrophic respiration was triple that on root respiration (37% vs. 12%). Therefore, warming studies that do not isolate the soil microbial respiration component would underestimate the soil microbial respiration response to warming (and thus underestimate the component of response that represents loss of stored soil C to the atmosphere). Clarifying the release of soil CO$_2$ from soil- or root-derived sources can help us fully understand the response mechanisms of soil carbon pool under climate warming.

Previous studies suggested that soil carbon emissions in grasslands had the highest uncertainty among all biomes[35,36]. In addition, Tibetan alpine grassland ecosystems, with huge carbon stocks[8] (7.4 Pg C in the top 1 m), may lose a large share of soil carbon with warming because warming increases thaw depth and duration[37], and the effect size may be contingent on the size of initial soil C stock (based on surface or shallow soil warming experiments[38]). But, thus far shallow warming experiments in alpine grassland ecosystems have not altered soil organic carbon stocks[20,39]. Therefore, it is crucial to clarify whether alpine grasslands will become carbon sources or carbon sinks under climate warming, considering the whole soil profile and plant inputs. In the first four years of our study, whole-

soil warming did not significantly affect soil organic carbon stocks, indicating that the elevated SOC-derived respiration was offset by an increase in plant biomass inputs to soil due to warming, or that the impact on SOC stock was not yet detectable. Our study also showed that the strength of the SOC-climate feedback may be underestimated for two reasons. First, because previous warming experiments missed the response of deeper soils to warming in grassland ecosystems. Second, most studies did not quantify the soil heterotrophic respiration response, which we showed was more sensitive to whole-soil warming compared to autotrophic respiration. Despite increased SOC-derived losses, it is difficult to significantly alter the vast soil carbon pool under relatively short-term warming (~3.3 years); whole-soil carbon content is resistant to early-stage climate warming in the alpine grassland.

In this study, we investigated the effects of four years of whole-soil warming on carbon dynamics in an alpine grassland ecosystem on the Tibetan Plateau. We found that whole-soil warming markedly increased total soil CO$_2$ efflux by 26%, which was much greater than the average effect size (7-8%) by experimental warming in the meta-analysis across global grasslands or alpine grasslands (none were whole-soil warming experiments). Moreover, the SOC-derived heterotrophic respiration increased by 37%, while the root-derived autotrophic respiration increased by 12% (but non-significant) by the whole-soil warming. Over the four years, the warming-induced relative increase in CO$_2$ effluxes did not change significantly. However, the large whole-soil SOC pool had not been significantly altered after ~3.3 years of warming. These results demonstrate that the expected future, which is long-term whole-soil warming, could create a much stronger effect on soil carbon emission than what is estimated by previous warming experiments that do not distinguish SOC- versus root-derived responses or that only warm surface soils.

## Methods

### Study site

This field warming experiment was situated in the Haibei National Field Research Station of Alpine Grassland Ecosystems on the northeast Tibetan Plateau, Menyuan county, Qinghai province, China (37°37′ N, 101°12′ E). Mean annual temperature (MAT) is -1.2 °C and mean annual precipitation is 487 mm with most of it occurring during the growing season (May to September, 5 months). This experiment is in an alpine meadow ecosystem at 3200 m above sea level. Alpine meadow is the main vegetation type on the Tibetan Plateau. The area of alpine meadow accounts for more than 44% of the area of alpine grasslands (including alpine meadow, alpine steppe, and swamp meadow), while its SOC storage accounts for 56% of the SOC storage of alpine grasslands on the whole Tibetan Plateau[8]. The alpine meadow is dominated by *Kobresia humilis*, *Stipa aliena*, *Elymus nutans*, *Gentiana straminea*, and *Tibetia himalaica*, and other herbaceous plants[40,41]. The soils in this area have a loam texture and are *Cryic Cambisols* (*Cambisols* is the second most extensive soil group on the Earth, occupying 12% of the total land area[42]) with a mean pH of 8.2[43] (Supplementary Table 2).

### Experimental design

This warming experiment warmed the soil 4 °C to 1 m depth (below 1 m is parent material; the global average soil thickness is about 1 m and the average soil thickness of alpine grasslands is about 0.7 m[44]) while maintaining the natural temperature gradient following the design of Pries et al. [6] and Hanson et al. [45]. Beginning on June 17, 2018. The whole-soil warming experiment consists of eight circular plots (four paired 'warm' and 'control' plots, 3 m diameter) (Supplementary Fig. 1). The minimum distance between plots is 2.5 m. Twenty 1-m long resistance heating cables (BriskHeat, Ohio, USA) were placed into stainless-steel rods and inserted vertically (then filled with sand), surrounding each plot at 0.5 m outside the plot perimeter. Two concentric rings of heating cables at 1 and 2 m in diameter were buried at 5 cm below the soil surface to compensate for surface heat loss. We also installed unheated cables in the control plots. The experiment was powered using a 220 V supply and the power to heat cables was routed through silicon-controlled rectifiers (SCRs, Watlow, Missouri, USA) controlled by a data logger (CR1000, Campbell Scientific, Utah, USA). A current-to-voltage converter (SDM-CVO4, Campbell Scientific, Utah, USA) connected to the CR1000 could send on-off signals under data logger program control to rectifiers. Power to deep heaters (1 m heating cables) was based on the temperature difference between paired control and heated temperature sensors at 30, 40, 60, 80, and 100 cm depth at a radial distance of 0.75 m from plot center, while power to surface heaters was based on the temperature difference between paired control and heated temperature sensors at 5, 10, and 20 cm depth at a radial distance of 0.75 m from plot center.

We used temperature sensors (thermistors customized by Lica United, Beijing, China) to monitor soil temperature at 5, 10, 20, 30, 40, 60, 80, and 100 cm depth at a radial distance of 0.75 m from plot center in all plots. We used moisture sensors (Delta-T, UK) to monitor soil volumetric water content at 10, 20, 30, 40, 60, and 100 cm depth at a radial distance of 0.75 m from plot center in all plots. Dataloggers continuously recorded soil temperature and moisture at 10 min intervals. Therefore, the power to the heaters could be adjusted every 10 min by the difference between paired control and warming plots to maintain a targeted 4 °C warming in the warming plots.

### Soil CO$_2$ efflux

Soil CO$_2$ efflux (respiration) was measured during the growing season (every two weeks from June 2018 until September 2020, approximately monthly from May 2021 until September 2021) within each plot using a chamber (with a 20 cm diameter) connected to an infrared gas analyzer. We used the Li-8100 automated soil CO$_2$ flux system (Licor, Lincoln, Nebraska, USA) until June 27, 2019, and then have been using

the PS-9000 automated soil CO$_2$ flux system (Lica United, Beijing, China) which is a Chinese-made equipment similar to the Li-8100. The CO$_2$ fluxes from these two equipments are very similar (with R$^2$ mostly >0.95) based on synchronous measurements during 2019. Soil CO$_2$ efflux was measured for five collars per plot of warming treatment (one root-exclusion and four root-ingrowth) and three collars per plot of control treatment (one root-exclusion and two root-ingrowth) to determine soil- and root-derived components of the soil CO$_2$ efflux. The root-exclusion collars were made from PVC tubing (20 cm diameter, 65 cm depth, 5 cm above soil surface), while the root-ingrowth collars were made from PVC tubing (20 cm diameter, 5 cm depth, 5 cm above soil surface). All these collars were placed in August 2016 to ensure a long enough time (22 months) to minimize the disturbance during installation on soil CO$_2$ efflux. Any new plants growing inside root-exclusion collars since previous measurement would be clipped before the next flux measurement. Therefore, the soil CO$_2$ efflux from root-exclusion collars is soil heterotrophic respiration (Rh, SOC-derived), and the soil CO$_2$ efflux from root-ingrowth collars is soil respiration (Rs, total). The difference between Rs and Rh is soil autotrophic respiration (Ra, root-derived). This method of separating SOC- and root-derived components of total soil respiration is widely used in the literature[13]. We also calculated the relative contributions of roots (root-derived) and heterotrophic microbes (SOC-derived) to total soil CO$_2$ efflux. Before the experimental treatments began, we had one sampling for soil CO$_2$ efflux. After the beginning of the experiment, there were 6 sampling dates in 2018, 10 each in 2019 and 2020, and 5 in 2021.

### Plant and soil properties

The peak above-ground biomass (four 0.25 × 0.25 m quadrats, randomly chosen within each plot) of the 8 plots (control and warming treatments, 4 replicates) were harvested at the end of August every year (2018, 2019, 2020, and 2021), oven-dried at 65 °C to constant weight and weighed. The harvested total plant dry matter (living biomass + dead detritus) was considered to be ANPP (aboveground net primary productivity), while the harvested living plant dry matter was AGB (aboveground biomass). Belowground net primary productivity (BNPP) was estimated by the root in-growth core method[46]. Near the end of the growing season (late August), one 40 cm depth soil core (5 cm diameter) was sampled. We removed all visible roots and then root-free soils were layered into nylon-mesh bags (5 cm diameter, 40 cm length, allowing roots to pass through) back into the hole. Then, at the same time (late August) in the following year, the bags were removed to pick out the newly grown roots, which were washed and dried at 65 °C to constant mass and weighed to obtain BNPP per unit area. Soil samples were collected from two soil cores per plot using a corer (5 cm diameter) from the soil profile (0-100 cm) at the end of August 2017 (before the warming treatments began, 0–10, 10–30, 30–50, 50–100 cm), August 2019 (14 months of warming, 0–10, 10–30, 30–60, 60–100 cm), and August 2021 (38 months of warming, 0–10, 10–30, 30–60, 60–100 cm). All fresh soil samples were stored at 4 °C using a cooler covered with ice-bags and taken back to the laboratory within 24 h. We discarded visible stones and picked roots when soil samples were sieved through 2 mm sieve. Live roots were also oven-dried at 65 °C to constant mass and weighed to obtain belowground biomass (BGB)[15]. The sieved soil samples were divided into three parts: one was air-dried for elemental analysis; one was stored at 4 °C for available nutrients, microbial biomass carbon (MBC), and nitrogen (MBN); and the other one was stored at -20 °C for enzyme activity analysis. Microbial carbon use efficiency (CUE) was also calculated based on microbial biomass and enzyme activity data using a stoichiometric method[13].

Soil pH was determined in the suspension (1:5, the ratio of soil to water) with a pH meter. Soil water content (SWC) was measured by calculating the mass loss with oven-drying at 105 °C for 48 h. Air-dried

soil samples were treated with 1 M hydrochloric acid (HCl) to remove carbonate and then washed to neutral with deionized water several times. Then soil organic carbon (SOC) and total nitrogen (TN) concentrations were determined with an Elemental Analyzer (Elementar vario, Langenselbold, Germany). The ammonium nitrogen ($NH_4^+$-N) and nitrate nitrogen ($NO_3^-$-N) were extracted with 2 M KCl solution (1:5, the ratio of soil to water) and the filtered solution was determined by a Continuous Flow Analyzer (AA3, Bran+Luebbe, Germany). The available nitrogen (AN) is the sum of $NH_4^+$-N and $NO_3^-$-N. Extractable organic carbon (EOC) and extractable total nitrogen (ETN) were extracted with 0.5 M $K_2SO_4$ solution (1:4, the ratio of soil to water) and the filtered solution was determined by a TOC analyzer (Multi N/C 3100, Analytik Jena, Germany). Soil MBC and MBN were measured by the chloroform-fumigation-extraction method with the extraction efficiency 0.45[47,48]. Soil extracellular enzyme activity was determined by the 96-well microplate method. Three soil hydrolytic enzymes involved in carbon (BG, β-1, 4-glucosidase), nitrogen (NAG, β-1, 4-N-acetyl-glucosaminidase), and phosphorous (AP, acid phosphatase) cycling were determined. The incubation time was 3 h and the Tris buffer (8.0) was close to the average pH of all soil samples. Then the fluorescence for hydrolytic enzymes was quantified by using a microplate reader (Synergy H1M, Biotek, USA). The determination method of enzyme activity was described in detail by Chen et al. [49].

## Statistical analyses

All the analyses were performed using the R platform [version 4.0.3][50]. To present the temporal patterns more clearly, we averaged the daily soil temperature and moisture data from the data logger (recorded once every 10 minutes, 144 records per day, Supplementary Figs. 2-3). The contributions of heterotrophic microbes (SOC-derived) and roots (root-derived) to total soil $CO_2$ efflux were calculated by Eqs. 1 and 2, respectively:

$$SOC - derived\ CO_2\ efflux\ (\%) = R_{ex}/R_{in} \times 100 \qquad (1)$$

$$Root - derived\ CO_2\ efflux\ (\%) = (R_{in} - R_{ex})/R_{in} \times 100 \qquad (2)$$

Where $R_{ex}$ and $R_{in}$ are the $CO_2$ efflux (μmol m$^{-2}$ s$^{-1}$) from root-exclusion and root-ingrowth collars, respectively.

The effects of warming on the percentage difference of mean annual soil $CO_2$ efflux (total, SOC-derived, root-derived) between warming and control treatments were calculated by Eq. 3:

$$\text{Warming response (\%increase in } CO_2 \text{ efflux)} \\ = (Warm_{Rx} - Control_{Rx})/Control_{Rx} \times 100 \qquad (3)$$

Where $Warm_{Rx}$ and $Control_{Rx}$ are the soil $CO_2$ efflux (μmol m$^{-2}$ s$^{-1}$, total, SOC-derived, root-derived) from the warming and control plots, respectively.

The repeated measures ANOVA was used to test the effects of warming treatment and time on soil $CO_2$ efflux (total, SOC-derived, root-derived), contributions of SOC-derived and root-derived components to total soil $CO_2$ efflux, and plant properties (ANPP, AGB, BNPP, and BGB) over four years (2018-2021). It was performed using the *ezANOVA* function of the "*ez*" R package and then the Bonferroni test was done to assess the effect of warming over time. Then we calculated the annual and total (4-year) average responses of soil $CO_2$ efflux to whole-soil warming. Moreover, we also calculated the corrected total average soil $CO_2$ efflux by the "*SpATS*" R package to reduce possible spatial heterogeneity[51,52]. We treated the warming treatment as a fixed effect and the rows of plots as random effects (because the columns of plots were already considered as potential random effects during the experimental design). The mixed-effects models were used to estimate the response of soil properties (SWC, SOC, $NH_4^+$-N, $NO_3^-$-N,

EOC, ETN, MBC, MBN, and CUE) to warming, with warming treatment, soil depth, and two-way interactions as fixed factors, and block nested within plot number as random effects. The linear mixed-effects models (lme) were fitted by the *lme* function of the "*nlme*" R package. A paired samples *t*-test was used to determine the effects of warming treatments on plant properties (ANPP, AGB, BNPP and BGB), mean soil $CO_2$ efflux, and percentage contributions (SOC-derived and root-derived). It was also used to estimate the warming effects on soil properties at different soil depths. Asterisks indicate statistically significant differences between warming and control treatment († $P < 0.10$, * $P < 0.05$, ** $P < 0.01$, *** $P < 0.001$).

## Meta-analysis for the warming experiment in global grasslands

To assess the general effect of experimental warming on soil respiration (total, SOC-derived, and root-derived) in global grasslands or alpine grasslands only (none were whole-soil warming), we collected the published papers before October 20, 2023 using the Web of Science (https://www.webofscience.com), China National Knowledge Infrastructure (https://www.cnki.net), and Google Scholar (https://scholar.google.com). The key words used for the article search were: (a) field experiment or manipulated experiment (excluding incubation experiments in the laboratory), (b) experimental warming or increased temperature or elevated temperature, (c) soil respiration or $CO_2$ flux, and (d) grassland or meadow or steppe. Publications selected for this meta-analysis had to meet the following criteria: (1) At least one of the considered three variables (total respiration, SOC-derived respiration, and root-derived respiration) was reported. (2) Studies included both control and warming treatments. (3) The means, standard deviations (SD) or standard errors (SE), and sample sizes of the selected variables were clearly reported or could be calculated from the data of publications. (4) The warming protocols (warming method, magnitude, and duration) were directly recorded. (5) Only the results from the ambient and warming treatments were used (e.g. excluding fertilization vs. fertilization and warming). Finally, 59 studies (from 72 articles) were included in this meta-analysis based on these criteria (Supplementary Fig. 15).

For categorical variables, warming methods were divided into infrared heater (IH) warming, open top chamber (OTC) warming, and other warming methods (including heating cable, infrared reflector, greenhouse, and translocation warming); warming magnitudes were categorized into low-level (<2 °C) and high-level (≥2 °C); experimental warming durations were categorized into short-term (<5 yr), middle-term (5-10 yr), and long-term (≥10 yr). Moreover, we also recorded a wide range of environmental variables related to warming experiments, including longitude, latitude, altitude, mean annual temperature (MAT) and annual precipitation (MAP). Note in this meta-analysis, we included multiple years results (rather than the latest year result) to obtain more observations, particularly for analyzing the effect of warming duration[53]. In the end, 234 observations were included in this study (Supplementary Fig. 16).

The effect size of experimental warming on each variable was quantified by calculating the natural logarithm of response ratios ($RR$)[54] (Eq. 4):

$$RR = \ln(X_T/X_c) \qquad (4)$$

Where $X_T$ and $X_C$ are the mean values of variables in warming and control treatments, respectively.

The variance ($v$) of each $RR$ was calculated by Eq. (5):

$$v = \frac{S_T^2}{N_T \bar{X}_T^2} + \frac{S_c^2}{N_c \bar{X}_c^2} \qquad (5)$$

with $N_T$ and $N_C$ as the sample sizes, $S_T$ and $S_C$ as the standard deviations of means in warming and control treatments, respectively.

The *rma.mv* function from the R package "*metafor*" was used to evaluate the weighted effect size and 95% confidence interval (CI) by random-effects models[15]. Observations from the same study were not independent because they came from different time points, therefore the "study" was treated as a random factor. The effects of warming were considered significant if the 95% CI did not overlap with zero. The $Q_B$ (Between-group heterogeneity) statistical test was used to compare the differences in weighted effect sizes among groups divided by the warming method. A significant $Q_B$ value ($P < 0.05$) suggested that the weighted effect sizes of a given variable differed among groups. The relative importance of predictors was expressed as the sum of Akaike weights for all models that included this predictor (based on mixed-effects meta-regression model), using the "*glmulti*" package in R[15]. A cutoff of 0.8 was set to differentiate between nonessential and important predictors[55]. The regression analyses were conducted to further show the relationships between RR of soil respiration (total data mean total soil respiration was measured, and paired data mean total soil respiration and its two components (root-derived and SOC-derived) were all measured) and experimental duration (which was identified to be a significant factor in Supplementary Fig. 14a). The database and articles list of this meta-analysis were deposited at https://github.com/yancypku/soil-respiration.

### Reporting summary

Further information on research design is available in the Nature Portfolio Reporting Summary linked to this article.

## Data availability

The data that support the findings of this study are openly available in figshare at https://doi.org/10.6084/m9.figshare.24921495.v2. Source data are provided with this paper.

## Code availability

The code that support the findings of this study are openly available in figshare at https://doi.org/10.6084/m9.figshare.24921495.v2.

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

## Acknowledgements

We thank the staff at the Haibei station of Chinese Academy of Sciences for providing logistical support in the field. We also sincerely thank all the scientists whose data were included in this meta-analysis. This study and B.Z. were financially supported by the National Key Research and Development Program of China (2022YFF0801902) and the National Natural Science Foundation of China (31988102 and 42141006).

## Author contributions

B.Z. conceived the idea; Y.C., W.K.Q., Q.F.Z., X.D.W., Y.H.H. and H.Y.Z. conducted this experiment, and collected and analyzed the data; J.G.F., M.G.H., Z.H.Z. and J.S.H. provided assistance in analyzing the data and maintaining the experiment; Y.C., B.Z. and M.S.T. wrote the manuscript with input from all authors.

## Competing interests

The authors declare no competing interests.
