## [Peer Review File · Nature Communications]

Whole-soil warming leads to substantial soil carbon emission in an alpine grasslandREVIEWER COMMENTS

Reviewer #1 (Remarks to the Author):

This manuscript describes a novel and interesting experiment on whole-soil warming at a study site on the Tibetan Plateau. In a field experiment, the soil temperature (0-100 cm depth) was increased to about 4°C over a period of four years. Several variables were measured to investigate the effects on CO₂ respiration and to determine whether the specific experiment would become a carbon source or sink given the increased temperature. The results showed that the warming did have an effect on the total CO₂ emission (by 25%) derived from soil organic carbon, but only minor derived from root CO₂ efflux. This suggests that future studies need to quantify soil microbial respiration as a component in the response to global warming effects. These results were compared in a meta-study. Suggesting that consideration of whole-soil warming will lead to a stronger soil organic carbon feedback during climate change.

The article addresses a very relevant and important issue that should urgently be studied. It is well written and presents a very interesting dataset and approach. However, the study presented was conducted at one experimental site over a relatively short period of time regarding the mechanisms studied (4 years). While I find the study and the experimental design very interesting, I am not sure if the results can be generalized and scaled up to a global level, as it is done at the end of this study, or if the study provides that big methodological insights for the whole community at this stage. My more detailed comments follow below:

The authors state that the problem studied is a global one, which I fully agree with. However, the present study only examines one experimental site over 4 years. In the discussion, the results of this study are used to calculate a global effect (L459 ff), without commenting on possible confounding factors. For example, how might the specific soil type affect these results? How might different altitudes affect the results? How can different soil microbial communities and/or plants affect the results? In my opinion, this comparison is exaggerated and should not be made without further explanation.

One would also expect the microbial and/or plant community to change over time as they adapt to the warmer temperature. This leads to two questions: What do the authors think about this and its influence on the results? And could this be tested in such a short experiment for such a complex mechanism by inoculating adapted soil microbiomes from other locations?

The manuscript describes the start of what I believe is an ongoing (longterm) experiment. However, the effect of the effective start is very interesting, but not specifically addressed in this manuscript. For example, in Figures S5 and S7d, you can see that the effect of the warming treatment becomes more pronounced in the later years.

I wondered why the authors did not try to use this very interesting and powerful data measured over time to analyze the effect of the time itself. Many of the variables are measured several times a year. So

instead of building a model and testing for significance within each time point, one could think of a model that integrates time as a variable and corrects for autocorrelation after stations. Instead of computing annual means and comparing them. I wonder if the authors have a specific reason for the proposed analysis.

The different plant related samples were all taken in different numbers and areas (AGB 4 samples per time at 0.25x0.25m, clipping was done on the whole plot (I assume), and soil samples were taken one core to 40cm depth per time point). Does this introduce some bias? For example, in Fig. S7 we can see a significant effect in litter, but no effect in AGB, which surprised me as one would expect a correlation between these traits.

Perhaps a rewrite and further elaboration of part 2.5 (especially L251-L263) would help a reader to get a deeper understanding of the analysis performed.

The authors have chosen a "delta" temperature increase, but in many regions the temperature change in the climate scenarios is expected to be seasonally pronounced. I think the methods/discussion section could benefit from more explanation of this choice and setup.

I would appreciate some more insight into the study site, e.g. soil type, depth, etc. Also, this is not put into a more global context, while the results are. I think this is a bit of an imbalance at the moment, which does not help the reader to understand and "categorize" the results.

Figures S2 and S3 show the temperature and SWC over time. However, the deviation is not shown in these figures. Are the values shown means/medians over the treatment or individual plots?

Also, in Figure S3 at 20cm and 30cm the soil moisture is higher for the control treatment throughout the measurement period. This is interesting because one could think that evaporation should be higher due to increased temperature and therefore soil moisture should be lower. Also, this layer is very important for the plants as a lot of processes are taking place, could this difference be influencing the results of the experiment?

Therefore, I wondered about the effect of the plot itself, or spatial heterogeneity. In the field I currently work in, this is a big issue, and we try to tackle it by statistically correcting for spatial heterogeneity. There is an easy-to-use R package called "SpATS" (<https://cran.r-project.org/web/packages/SpATS/SpATS.pdf>) that you might be interested in using in this study to reduce spatial effects and thus improve the ability to find and describe the underlying dispersal mechanisms. With this package, one can fit two-dimensional P-Splines in combination with a linear mixed effects model through the experiment by providing the spatial information and hence correct for spatial trends.

I liked the additional meta-analysis, which helped the reader to understand and evaluate the results of the "main" study. The discussion uses this very well by defining three different main sources on different

levels. However, I miss a part of the discussion where the limitations of this study are discussed in detail. Also, for me as a reader, it would be more intuitive if the discussion started with the experiment itself and its results, and then broadened the scope. I would like to see this section restructured.

In Figure S14, the authors show the response ratios of soil respiration with respect to the duration of the experiment. The deviation between the different studies seems to be very large. Hypocritically, why is this study not an outlier or one of the very far outliers? In my opinion, it would strengthen the study if the authors provided some further explanations.

Minor:

Line 257: abbreviation: lme is not introduced

In line 404 you mentioned OTC, this is just defined in the Supplementary material, but never in the main manuscript.

Figure 5 and Table 1 are mentioned the first time in the discussion, please include them in the Results section.

Line 347 ff: I do not really get the point the authors want to make with the different statements, as they contradict. Maybe add another sentence to bring that in perspective.

Figure 2: Nice plot, however, the x-axis description could profit if its changed to just the names of the month instead of always writing the first day as well. (same for Fig S5)

Figure 4. It is hard to see which error bar belongs to which point, please add a little offset between them (or jitter in ggplot).

Reviewer #2 (Remarks to the Author):

This manuscript describes a study that is timely and has yielded interesting results by showing that in a whole-soil warming experiment in the Tibetan alpine grassland, warming-induced increases in heterotrophic soil respiration exceeded those of autotrophic (and total) soil respiration. The authors compare their experimental results (with four years of data) to those of a meta-analysis of warming effects on soil respiration in global grasslands (from 74 papers). Then, they concluded that "whole soil warming causes a much stronger SOC-climate feedback in the alpine grassland ecosystem than what was reported in previous warming experiments, most of which only heat surface soils".

Overall, the experimental design is reasonable, the results and conclusions are solid and novel, and the writing is clear. This work provides a new viewpoint to estimate the carbon-climate feedback, and makes a strong contribution to the literature of soil biogeochemistry and ecosystem ecology.

I have a few specific comments below for authors to consider during revision, and hope these comments would help the authors improve the manuscript before its publication.

1. The authors only measured soil respiration fluxes during the growing season (from May to October). Although the carbon fluxes have been shown to be very low in the non-growing season of the alpine grassland (line 246-249), I wonder whether the authors have data from the non-growing season for this experiment. I understand that the conditions are very harsh in the winter and the fluxes are very low compared to those during the growing season. But, are they also sensitive to the warming (which is year-round)?
2. The results of the soil C budget (inputs and outputs, Table 1) are very interesting, which are novel compared to other whole-soil-warming experiments in forests. Because the experimental duration is only four years and the effect size is relatively small (0.82% per year of total 0-100 cm SOC stock, line 448-453), the SOC had not been significantly changed by the whole-soil-warming over the ~4 years. Can the authors do a power-analysis and estimate how long (years) would it take to observe a statistically significant change (decrease) in the whole-soil (0-100 cm) SOC stock if the effect size continues?
3. For the meta-analysis, can the authors show a prisma diagram (<http://www.prisma-statement.org>)? It would make the results more transparent. A map showing the spatial distribution of the sites for these warming experiments is also helpful. Moreover, it is surprising to me that the warming magnitude is not an important mediator of the warming effect on soil respiration fluxes (Fig. S13, supplementary materials). Can the authors discuss the possible reasons for these results, although a warming study with different levels of warming magnitude is better to answer this question (instead of a meta-analysis which includes results from experiments with different magnitudes, durations, methods, ecosystems, ...).

Reviewer #3 (Remarks to the Author):

Overall, this is a timely manuscript in which the authors sought to determine how whole-soil (down to 1 meter depth) warming affects carbon cycling in an arctic grassland ecosystem. They also conducted a meta-analysis of warming experiments in grasslands around the globe, but there are many details missing and the presentation of the meta-analysis approach and results are not well integrated into the manuscript.

The field experiment is robust and I have no concerns with how it is justified, described, or presented. My biggest concern with this paper is the confusing aspect of the meta-analysis. It is mentioned in the Abstract, Methods, and Supplement, but not the Introduction, Results, or Discussion. It is mentioned as one sentence in the Conclusions section. Where they do describe it, significant details are missing, making it not possible to evaluate the strength of their results. While important, the meta-analysis feels like an “add-on” to the paper, rather than being fully integrated into it. I highly recommend the authors move some of the key aspects of the meta-analysis into the main body of the paper since most readers don't go to the Supplement or remove it from their paper.

The following detailed comments are meant to improve the manuscript:

Title and lines 31-32: The authors use the title to summarize “whole-soil warming” and explain that they are the first to do this in an alpine grassland ecosystem, but then they introduce results from a meta-analysis across global grasslands here. I suggest they clarify up front whether the manuscript is also about several arctic grasslands around the globe or just one (their experiment) by either changing the title and text in line 28 or making it clearer in lines 31-32 what studies are included in the meta-analysis that are not a “whole soil warming experiment.”

Related to above comment: The entire paragraph in lines 85-93 only mentions the warming experiment, but not a meta-analysis. I suggest having the Abstract and this paragraph match up in terms of what this paper is about. Since the authors conducted a meta-analysis, that should be included in all aspects of the paper: Title, Introduction, etc.,

The abstract is a bit confusing in regarding to warming-induced effects on SOC. These statements are all made:

-“...whole-soil warming stimulated total and SOC-derived CO₂ efflux by 26% and 37%, respectively”

-“However, the whole-soil SOC pool had not been significantly altered in this study due to the relatively short warming duration (4 years).”

-“We show that whole-soil warming causes a much stronger SOC-climate feedback in the alpine grassland ecosystem than what was reported in previous warming experiments,...”

It is understandable that CO₂ will respond more rapidly compared to bulk soil SOC pools, but it is unclear how the authors conclude the last sentence about SOC-climate feedbacks. What evidence are they using to draw this conclusion?

Lines 265-279: There are many important details missing from the description of their meta-analysis methods that are standard when using meta-analysis techniques:

-How many total papers did the Web of Science search bring up?

When they say there were 234 observations, does this mean their search turned up 234 papers, but they used only 72 or that there were 234 observations across the final 72 papers they used? Based on the figures it looks like there were 234 observations (not papers), but this needs to be clearer in the Methods and Results sections.

-How did they treat papers that had multiple time points for soil respiration? I see in Fig. S11 that studies were grouped into three time ranges: < 5, 5-10, and >10 years. Did they average within each paper for these three separate time periods?

-Did they include both lab and field studies? I see they only included experiments, but did these include lab experiments with soils collected from grasslands?

-Were the studies in the meta-analysis constrained to “whole-soil warming experiments” only or no? If not, how are they comparing their “whole-soil warming in an alpine grassland” to “any depth of soil warming to grasslands around the globe”?

-Related to above, did the meta-analysis include warming experiments in alpine grasslands? This would presumably be in a non “whole-soil warming experiment” since theirs is the first, but having a comparison between their experiment and other warming experiments in alpine grasslands would strengthen their paper.

-The field experiment is about “arctic grasslands”, yet the meta-analysis is about grasslands around the globe at all latitudes. It’s unclear why they didn’t match their analyses to the same scale. Since the results of the meta-analysis include grasslands across a range of latitudes, I don’t think it’s fair to compare the 26% loss of SOC-derived CO₂ to 9% found in their grasslands that they mention in the Abstract. I think it’s ok to include this comparison, but then they should also explain what latitudes these values come from, make it clear whether the meta-analysis includes soils that weren’t warmed to such depths, or the comparison it between apples and oranges.

-S12: The 95% confidence intervals across all studies should be shown.

Without this information it is not possible to evaluate the results of their meta-analysis.

The authors need to include a table of all 234 studies they included in their meta-analysis.

Results and Discussion: there is no mention of the results of the meta-analysis. It’s unclear why this is missing in these important sections.

Reviewer(s)' Comments to Author:

Reviewer #1 (Remarks to the Author):

This manuscript describes a novel and interesting experiment on whole-soil warming at a study site on the Tibetan Plateau. In a field experiment, the soil temperature (0-100 cm depth) was increased to about 4 °C over a period of four years. Several variables were measured to investigate the effects on CO₂ respiration and to determine whether the specific experiment would become a carbon source or sink given the increased temperature. The results showed that the warming did have an effect on the total CO₂ emission (by 25%) derived from soil organic carbon, but only minor derived from root CO₂ efflux. This suggests that future studies need to quantify soil microbial respiration as a component in the response to global warming effects. These results were compared in a meta-study. Suggesting that consideration of whole-soil warming will lead to a stronger soil organic carbon feedback during climate change.

[Response] Thanks for your constructive comments and insightful suggestions which greatly improved our work. Our detailed responses to your comments are listed below (line numbers refer to the clean version without track changes).

The article addresses a very relevant and important issue that should urgently be studied. It is well written and presents a very interesting dataset and approach. However, the study presented was conducted at one experimental site over a relatively short period of time regarding the mechanisms studied (4 years). While I find the study and the experimental design very interesting, I am not sure if the results can be generalized and scaled up to a global level, as it is done at the end of this study, or if the study provides that big methodological insights for the whole community at this stage. My more detailed comments follow below:

[Response] Thanks for your constructive comments.

1. To explore the potential global significance of our experiment in the Tibetan alpine grassland, we simply scaled up our main finding to a global level: **a**). The net loss of soil C stock is 180.7 g C m⁻² yr⁻¹ (Table 1) and the global grassland area is 52.2 × 10⁶ km² (Li et al. 2020). Moreover, the anthropogenic fossil CO₂ emissions in the last decade (2010–2019) is 9.6 ± 0.5 Pg C yr⁻¹ (Friedlingstein et al. 2020). **b**). Therefore, simply multiplying the annual net loss of soil carbon per unit area by the global grassland area, we could

obtain the total annual net loss of soil carbon from global grasslands (estimated annual net SOC loss of global grassland is 9.44 Pg C yr⁻¹). c). So to put it simply, if the effect of whole-soil warming continues, the net carbon loss by global grasslands is approximately equal to the fossil CO₂ emissions each year (which surely has large uncertainty and needs further investigation). We also modified the sentences to make it easier to understand how to scale up our main finding (line 290-295) as following:

“To explore the potential global significance of our experiment in the Tibetan alpine grassland, if we simply scale our main finding (loss of 180.7 g C m⁻² per year) to all global grasslands (52.2 × 10⁶ km², ref. 41), the estimated annual net SOC loss of global grasslands (simply multiplying the annual net loss of soil carbon per unit area by the global grassland area) would be 9.44 Pg C per year, a response comparable to fossil CO₂ emission in the last decade (2010–2019) (9.6 ± 0.5 Pg C yr⁻¹, ref. 2).”

2. Our extrapolated results may not be generalizable, because these were simple calculations which did not consider other confounding factors and site-to-site variations at the global level. The results of this study were only from a grassland ecosystem, and there were only three similar experiments in forests (Supplementary Table 1). Therefore, we call for more studies from different ecosystems among different sites to verify and generalize our conclusions in the future. Hopefully, after publication of this work, the scientific community will pay more attention to whole-soil warming experiments and a coordinated distributed experimental network of such experiments is emerging. **If you are not satisfied with this simple scale-up calculation, we can also delete this part.**

Friedlingstein P, O'Sullivan M, Jones MW, Andrew RM, Hauck J, Olsen A, Peters GP, Peters W, Pongratz J, Sitch S, Le Quere C, Canadell JG, Ciais P, Jackson RB, Alin S, Aragao LEOC, Arneeth A, Arora V, Bates NR, Becker M, Benoit-Cattin A, Bittig HC, Bopp L, Bultan S, Chandra N, Chevallier F, Chini LP, Evans W, Florentie L, Forster PM, Gasser T, Gehlen M, Gilfillan D, Gkritzalis T, Gregor L, Gruber N, Harris I, Hartung K, Haverd V, Houghton RA, Ilyina T, Jain AK, Joetzjer E, Kadono K, Kato E, Kitidis V, Korsbakken JI, Landschutzer P, Lefevre N, Lenton A, Lienert S, Liu Z, Lombardozzi D, Marland G, Metzl N, Munro DR, Nabel JEMS, Nakaoka SI, Niwa Y, O'Brien K, Ono T, Palmer PI, Pierrot D, Poulter B, Resplandy L, Robertson E, Rodenbeck C, Schwinger J, Seferian R, Skjelvan I, Smith AJP, Sutton AJ, Tanhua T, Tans PP, Tian H, Tilbrook B,

Van der Werf G, Vuichard N, Walker AP, Wanninkhof R, Watson AJ, Willis D, Wiltshire AJ, Yuan WP, Yue X, Zaehle S (2020) Global Carbon Budget 2020. *Earth System Science Data* 12: 3269-3340.

Li L, Chen J, Han XG, Zhang W, Shao C (2020) Grassland ecosystems of China: A synthesis and resume. Springer Singapore.

The authors state that the problem studied is a global one, which I fully agree with. However, the present study only examines one experimental site over 4 years. In the discussion, the results of this study are used to calculate a global effect (L459 ff), without commenting on possible confounding factors.

For example, how might the specific soil type affect these results? How might different altitudes affect the results? How can different soil microbial communities and/or plants affect the results?

In my opinion, this comparison is exaggerated and should not be made without further explanation.

[Response] Thanks for your insightful suggestions. We added some discussion on the possible confounding factors (line 213-219) as following:

“Notably, our results that whole-soil warming significantly elevated soil CO₂ efflux were obtained from only one whole-soil warming experiment in an alpine grassland which cannot represent global grassland ecosystems. The response of soil CO₂ efflux to whole-soil warming may vary depending on environmental factors, soil types, plant and soil microbial communities, which requires verification by future coordinated distributed experiments with similar designs in other ecosystems and sites.”

One would also expect the microbial and/or plant community to change over time as they adapt to the warmer temperature. This leads to two questions: What do the authors think about this and its influence on the results? And could this be tested in such a short experiment for such a complex mechanism by inoculating adapted soil microbiomes from other locations?

[Response] Thanks for your insightful comments and suggestions.

1. We agree that plant or soil microbial communities would change over time as they adapt to higher temperatures. For example, the long-term warming (26 years, Harvard Forest) effects on soil respiration changed with time as microbial communities reorganized (Melillo et al. 2017; Metcalfe 2017). Our whole-soil warming experiment in the alpine

grassland, although had been going on for four years (from June 2018 to September 2021), is still relatively short. We plan to continue it for another 8-10 years in the future to see how plant and soil microbial communities adapt to the warming and how they affect ecosystem carbon cycling.

2. Your idea is ingenious, by inoculating adapted soil microbiomes from other locations to our site to verify the complex mechanism. We are very open to collaborations with any interested colleagues (including you) on this work in the future.

We also discussed the necessity for a long-term warming response (line 234-235) as following:

“Therefore, the long-term patterns of soil CO₂ emissions and soil C stocks in response to warming need to be further investigated¹².”

Melillo JM, Frey SD, DeAngelis KM, Werner WJ, Bernard MJ, Bowles FP, Pold G, Knorr MA, Grandy AS (2017) Long-term pattern and magnitude of soil carbon feedback to the climate system in a warming world. *Science* 358: 101-105.

Metcalf DB (2017) Microbial change in warming soils. *Science* 358: 41-42.

The manuscript describes the start of what I believe is an ongoing (longterm) experiment. However, the effect of the effective start is very interesting, but not specifically addressed in this manuscript. For example, in Figures S5 and S7d, you can see that the effect of the warming treatment becomes more pronounced in the later years.

[Response] Thanks for your insightful comments and careful observations. We also discussed the necessity for a long-term warming response (line 234-235) as following:

“Therefore, the long-term patterns of soil CO₂ emissions and soil C stocks in response to warming need to be further investigated¹².”

I wondered why the authors did not try to use this very interesting and powerful data measured over time to analyze the effect of the time itself. Many of the variables are measured several times a year. So instead of building a model and testing for significance within each time point, one could think of a model that integrates time as a variable and corrects for autocorrelation after stations. Instead of computing annual means and comparing

them. I wonder if the authors have a specific reason for the proposed analysis.

[Response] Thanks for your constructive comments.

1. To eliminate the influence of repeated measurements on the responses of soil CO₂ efflux (reducing autocorrelation), the repeated measures ANOVA was used to test the effects of warming and time (Fig. 2).
2. Moreover, instead of testing the significance within each time point (Fig. 2), we compared the warming effects on annual mean soil CO₂ efflux (Supplementary Fig. 8). This is a common method for presenting multi-year results in detail and has been used in many similar studies (Wan et al. 2009; Sharkhuu et al. 2016; Soong et al. 2021).

Sharkhuu A, Plante AF, Enkhmandal O, Gonneau C, Casper BB, Boldgiv B, Petraitis PS (2016) Soil and ecosystem respiration responses to grazing, watering and experimental warming chamber treatments across topographical gradients in northern Mongolia. *Geoderma* 269: 91-98.

Soong JL, Castanha C, Pries CEH, Ofiti N, Porras RC, Riley WJ, Schmidt MWI, Torn MS (2021) Five years of whole-soil warming led to loss of subsoil carbon stocks and increased CO₂ efflux. *Science Advances* 7: eabd1343.

Wan SQ, Xia JY, Liu WX, Niu SL (2009) Photosynthetic overcompensation under nocturnal warming enhances grassland carbon sequestration. *Ecology* 90: 2700-2710.

The different plant related samples were all taken in different numbers and areas (AGB 4 samples per time at 0.25x0.25m, clipping was done on the whole plot (I assume), and soil samples were taken one core to 40cm depth per time point). Does this introduce some bias? For example, in Fig. S7 we can see a significant effect in litter, but no effect in ABG, which surprised me as one would expect a correlation between these traits.

[Response] Thanks for your constructive comments and careful reading.

1. Aboveground biomass (in August) and clipped litter (in December) were all harvested by using four 0.25 × 0.25 m quadrats (randomly chosen with each plot) among the 8 plots (Fig. SS1). We did not clip the whole plot until the clipped litter was measured. Soil samples (after biomass harvest in August) were collected from two soil cores (0-100 cm, 5-cm diameter) within each plot. Belowground net primary productivity was estimated by one root in-growth core (0-40 cm, 5-cm diameter) per plot. More cores (soils and roots) would cause too much disturbance to these plots. These were all used to reduce possible

bias. We also revised the description of the sampling section to reduce possible misunderstandings (line 397-398 and 405-409) as following:

“Clipped litter was measured at the end of December every year (2018, 2019, 2020, and 2021, using the same method with AGB), and then all plots were clipped to simulate grazing.”

“Soil samples were collected from two soil cores per plot using a corer (5 cm diameter) from the soil profile (0-100 cm) at the end of August 2017 (before the warming treatments began, 0–10, 10–30, 30–50, 50–100 cm), August 2019 (14 months of warming, 0–10, 10–30, 30–60, 60–100 cm), and August 2021 (38 months of warming, 0–10, 10–30, 30–60, 60–100 cm)”.

2. The warming increased clipped litter only significantly in the last two years, while warming also increased AGB significantly in the first year (Supplementary Fig. 4). We added a regression figure to show the relationships between AGB and litter over four years (2018-2021, Fig. SS2). We found a positive relationship between these two variables in 2019 and 2020, although neither of these relationships were statistically significant. Therefore, in future work, we will study the response of plant communities to whole-soil warming in more detail (e.g. sampling bigger area and using more quadrats for each plot).

Fig. SS1 Schematic diagram of aboveground biomass (AGB) and clipped litter harvesting method. Both AGB and clipped litter were harvested by using four 0.25×0.25 m quadrats (randomly chosen within each plot).

Fig. SS2 The relationships between aboveground biomass and clipped litter over four years (2018-2021).

Perhaps a rewrite and further elaboration of part 2.5 (especially L251-L263) would help a reader to get a deeper understanding of the analysis performed.

[Response] Thanks for your insightful suggestions. We restructured this part to make it easier for readers to read (line 475-489) as following:

“The repeated measures ANOVA was used to test the effects of warming treatment and time on soil CO₂ efflux (total, SOC-derived, root-derived) and contributions of SOC-derived and root-derived components to total soil CO₂ efflux, and it was performed using the *ezANOVA* function of the “*ez*” R package. Then we calculated the annual and total (4-year) average responses of soil CO₂ efflux to whole-soil warming. Moreover, we also calculated the corrected total average soil CO₂ efflux by the “*SpATS*” R package to reduce possible spatial heterogeneity^{54,55}. The mixed-effects models were used to estimate the response of soil properties (SWC, SOC, NH₄⁺-N, NO₃⁻-N, EOC, ETN, MBC, MBN, and CUE) to warming, with warming treatment, soil depth, and two-way interactions as fixed factors, and block nested within plot number as random effects. The linear mixed-effects models (*lme*) were fitted by the *lme* function of the “*nlme*” R package. A paired samples *t*-test was used to determine the effects of warming treatments on plant properties (AGB, BGB, BNPP, and clipped litter), mean soil CO₂ efflux, and percentage contributions (SOC-derived and root-derived). It was also used to estimate the warming effects on soil properties at different soil depths. Asterisks indicate statistically significant differences between warming and control treatment († $P < 0.10$, * $P < 0.05$, ** $P < 0.01$, *** $P < 0.001$).”

The authors have chosen a "delta" temperature increase, but in many regions the temperature

change in the climate scenarios is expected to be seasonally pronounced. I think the methods/discussion section could benefit from more explanation of this choice and setup.

[Response] Thanks for your constructive comments. We fully agree with you that there may be seasonal fluctuations in the temperature increase in different regions. However, due to limited funding and being consistent with other whole-soil warming experiments (e.g. Pries et al. 2017, Nottingham et al. 2020), we could only take a year-round constant temperature increase (4 °C). We added relevant discussions (line 244-247) as following:

“Finally, we used a year-round constant temperature increase (4 °C) in this study, but there would be seasonal fluctuations in the temperature increase in different regions³⁴. Therefore, we need to set up seasonal asymmetric warming experiments to study the response of ecosystems in the future.”

Nottingham AT, Meir P, Velasquez E, Turner BL (2020) Soil carbon loss by experimental warming in a tropical forest. *Nature* 584: 234-237.

Pries CEH, Castanha C, Porras RC, Torn MS (2017) The whole-soil carbon flux in response to warming. *Science* 355: 1420-1422.

I would appreciate some more insight into the study site, e.g. soil type, depth, etc. Also, this is not put into a more global context, while the results are. I think this is a bit of an imbalance at the moment, which does not help the reader to understand and "categorize" the results.

[Response] Thanks for your constructive comments.

1. The soil type of *Cambisols* is the second most extensive soil group (order) on the Earth, occupying 12% of the total continental land area (Zech et al. 2022).
2. The global average soil thickness is about 1 m (Liu et al. 2023). Moreover, the reference soil depth of all soil units is also set at 1 m in the Harmonized World Soil Database (HWSD V1.2, global) (FAO 2012).

Therefore, our results are representative. We also added these details into the method section (line 332-334 and 337-340) as following:

“The soils in this area have a loam texture and are *Cryic Cambisols* (*Cambisols* is the second most extensive soil group on the Earth, occupying 12% of the total land area, ref. 44) with a mean pH of 8.2 (ref. 45, Supplementary Table 2).”

“This warming experiment warmed the soil 4 °C to 1 m depth (below 1 m is parent material; the global average soil thickness is about 1 m and the reference soil depth is also set at 1 m in the Harmonized World Soil Database, ref. 46) while maintaining the natural temperature gradient following the design of ref. 6 and ref. 47, beginning on June 17, 2018.”

FAO/IIASA/ISRIC/ISS-CAS/JRC (2012) Harmonized World Soil Database (version 1.2).
FAO, Rome, Italy.

Liu JT, Zhao W, Liu Y (2023) Modelling soil thickness evolution: Advancements and Challenges. *Acta Pedologica Sinica*.

<https://kns.cnki.net/kcms/detail/32.1119.p.20230420.0937.002.html>

Zech W, Schad P, Hintermaier-Erhard G (2022) *Soils of the world*. Springer Berlin.

Figures S2 and S3 show the temperature and SWC over time. However, the deviation is not shown in these figures. Are the values shown means/medians over the treatment or individual plots?

[Response] We revised Supplementary Figs. 2-3 (the values shown means over the treatment, n = 4) and added the standard error.

Also, in Figure S3 at 20cm and 30cm the soil moisture is higher for the control treatment throughout the measurement period. This is interesting because one could think that evaporation should be higher due to increased temperature and therefore soil moisture should be lower. Also, this layer is very important for the plants as a lot of processes are taking place, could this difference be influencing the results of the experiment?

[Response] Thanks for your constructive comments. We revised Supplementary Fig. 3 and added the standard error. Neither volumetric water content (continuously measured, highly variable among plots and layers, Supplementary Fig. 3) nor gravimetric water content (measured during destructive soil samplings in August 2019 and 2021, Fig. 1) responded significantly to whole-soil warming in this alpine grassland. Therefore, we can conclude that the warming effect on soil moisture across the 0-100 cm soil profile is generally minor, despite plot-to-plot and time-to-time variations.

Therefore, I wondered about the effect of the plot itself, or spatial heterogeneity. In the field I currently work in, this is a big issue, and we try to tackle it by statistically correcting for

spatial heterogeneity. There is an easy-to-use R package called "SpATS" (<https://cran.r-project.org/web/packages/SpATS/SpATS.pdf>) that you might be interested in using in this study to reduce spatial effects and thus improve the ability to find and describe the underlying dispersal mechanisms. With this package, one can fit two-dimensional P-Splines in combination with a linear mixed effects model through the experiment by providing the spatial information and hence correct for spatial trends.

[Response] Thanks for your insightful suggestions. According to your suggestion, we added a new figure (Supplementary Fig. 9) similar to Fig. 3b, but this figure used the “*SpATS*” R package to reduce the effect of spatial heterogeneity on soil CO₂ efflux. The response of total soil CO₂ efflux to whole-soil warming after correction is similar to that of non-correction (23% vs. 26%), but there was little difference between SOC-derived and root-derived CO₂ efflux in response to warming before and after correction (Fig. 3b and Supplementary Fig. 9). We also added these results into the results section (line 138-142) as following:

“Moreover, the response of total soil CO₂ efflux to whole-soil warming after correction (to reduce spatial heterogeneity) was similar to that of non-correction (23% vs. 26%), and there was little difference between SOC-derived and root-derived CO₂ efflux in response to warming before and after correction (Fig. 3b and Supplementary Fig. 9).”

Supplementary Figure 9. The mean soil CO₂ efflux (corrected value) partitioned into SOC-derived (heterotrophic respiration) and root-derived (autotrophic respiration) components for growing season measurements in four years of warming periods (from June 2018 to September 2021). The soil CO₂ efflux was corrected by the “*SpATS*” R package to reduce spatial heterogeneity. Different colors mean different treatments (blue indicates control and red indicates warming). Statistical significance of differences between control and warming treatment is shown by asterisks († $P < 0.10$, * $P < 0.05$, ** $P < 0.01$, *** $P < 0.001$, $n = 4$) or as non-significant (ns).

I liked the additional meta-analysis, which helped the reader to understand and evaluate the results of the "main" study. The discussion uses this very well by defining three different main sources on different levels. However, I miss a part of the discussion where the limitations of this study are discussed in detail. Also, for me as a reader, it would be more intuitive if the discussion started with the experiment itself and its results, and then broadened the scope. I would like to see this section restructured.

[Response] Thanks for your insightful suggestions. We restructured the discussion to make it easier for readers to read. Originally, we wanted to include a subtitle in the discussion section, but this is not allowed in this journal. Therefore:

1. [Paragraphs 1-2 of the discussion section]. This part summarized our main findings from the field experiment and compared them with other studies, and then investigated the possible mechanisms.
2. [Paragraphs 3-4 of the discussion section]. This part contained the limitations of this study.
3. [Paragraphs 5-6 of the discussion section]. This part contained appropriate extensions of our main findings.
4. [Paragraph 7 of the discussion section]. This part is the conclusion of this study.

In Figure S14, the authors show the response ratios of soil respiration with respect to the

duration of the experiment. The deviation between the different studies seems to be very large. Hypocritically, why is this study not an outlier or one of the very far outliers? In my opinion, it would strengthen the study if the authors provided some further explanations.

[Response] Thanks for your constructive comments. This meta-analysis did not include our results from whole-soil warming, nor any other results from whole-soil warming or whole-ecosystem warming (Supplementary Table 3). Whether our result is one of the outliers (Supplementary Fig. 14) will take a longer time to verify. We also discussed it (line 234-235) as following:

“Therefore, the long-term patterns of soil CO₂ emissions and soil C stocks in response to warming need to be further investigated¹².”

Minor:

Line 257: abbreviation: lme is not introduced

[Response] Done.

In line 404 you mentioned OTC, this is just defined in the Supplementary material, but never in the main manuscript.

[Response] Done. According to another reviewer's suggestion, we moved the detailed method of our global-scale meta-analysis from the supplementary materials to the main text of the manuscript.

Figure 5 and Table 1 are mentioned the first time in the discussion, please include them in the Results section.

[Response] Done.

Line 347 ff: I do not really get the point the authors want to make with the different statements, as they contradict. Maybe add another sentence to bring that in perspective.

[Response] Done. We added a sentence to reduce possible misunderstandings (line 176-178) as following:

“Prior to these whole-soil warming experiments, most warming experiments could only warm the surface soils (0-20 cm), while the effects of previous warming on soil respiration were diverse.”

Figure 2: Nice plot, however, the x-axis description could profit if its changed to just the names of the month instead of always writing the first day as well. (same for Fig S5)

[Response] Done. We have redrawn these two figures.

Figure 4. It is hard to see which error bar belongs to which point, please add a little offset between them (or jitter in ggplot).

[Response] Done.

Overall, we thank you sincerely for providing these constructive comments and insightful suggestions which greatly improved this work. We hope our revisions and clarifications have satisfactorily addressed your concerns and comments.

Reviewer #2 (Remarks to the Author):

This manuscript describes a study that is timely and has yielded interesting results by showing that in a whole-soil warming experiment in the Tibetan alpine grassland, warming-induced increases in heterotrophic soil respiration exceeded those of autotrophic (and total) soil respiration. The authors compare their experimental results (with four years of data) to those of a meta-analysis of warming effects on soil respiration in global grasslands (from 74 papers). Then, they concluded that "whole soil warming causes a much stronger SOC-climate feedback in the alpine grassland ecosystem than what was reported in previous warming experiments, most of which only heat surface soils".

Overall, the experimental design is reasonable, the results and conclusions are solid and novel, and the writing is clear. This work provides a new viewpoint to estimate the carbon-climate feedback, and makes a strong contribution to the literature of soil biogeochemistry and ecosystem ecology.

I have a few specific comments below for authors to consider during revision, and hope these comments would help the authors improve the manuscript before its publication.

[Response] Thanks for your constructive comments and insightful suggestions which improved our work. Our detailed responses to your comments are listed below (line numbers refer to the clean version without track changes).

1. The authors only measured soil respiration fluxes during the growing season (from May to October). Although the carbon fluxes have been shown to be very low in the non-growing season of the alpine grassland (line 246-249), I wonder whether the authors have data from the non-growing season for this experiment. I understand that the conditions are very harsh in the winter and the fluxes are very low compared to those during the growing season. But, are they also sensitive to the warming (which is year-round)?

[Response] Thanks for your constructive comments. Non-growing season carbon fluxes are considered very low at our site, so soil CO₂ efflux of non-growing season was not measured in the early three years (2018-2020). Currently, we have the data on soil CO₂ efflux (soil respiration) for the period between October 2020 and April 2021 (non-growing season, five measurements) (Fig. SS3). The analysis of Fig. SS3 was in complete agreement with Fig. 2. The mean value of soil CO₂ efflux in our study in the non-growing season was 17.5% of that

in the growing season (the mean value in the non-growing season was $0.53 \mu\text{mol m}^{-2} \text{s}^{-1}$, and the mean value in the growing season was $3.02 \mu\text{mol m}^{-2} \text{s}^{-1}$). The whole-soil warming of the non-growing season increased Rs (soil respiration, total, **significant**) and Rh (soil heterotrophic respiration, SOC-derived, **marginally significant**), while Ra (soil autotrophic respiration, root-derived) did not respond to warming (Fig. SS3). **The response of soil CO₂ efflux to whole-soil warming in the non-growing season was similar to that in the growing season, and both were sensitive to whole-soil warming.** However, the current amount of data on soil CO₂ efflux in the non-growing season is too small, and we may focus on the response of non-growing season soil CO₂ efflux to whole-soil warming in future studies. For example, we plan to apply for funding to buy automatic chambers (Li-8150) to continuously (every 30 minutes) measure soil CO₂ efflux from these 8 plots all year round.

Fig. SS3 Soil CO₂ efflux from control and warming treatments between October 2020 and April 2021 (non-growing season). (a) Rs (soil respiration, total). (b) Rh (soil heterotrophic respiration, SOC-derived). (c) Ra (soil autotrophic respiration, root-derived). During this non-growing season, there were five measurements, three in 2020 (October 16, November 20, and December 16), and two in 2021 (March 21 and April 22). Note that we did not measure it during the two coldest months (January and February) and these mean flux values may over-estimate the mean values during the whole non-growing season. The orange shaded areas represent warming periods. Different colors mean different treatments (blue indicates control and red indicates warming). Error bars are standard errors ($n = 4$). Statistical significance is shown by asterisks ($\dagger P < 0.10$, $* P < 0.05$, $** P < 0.01$, $*** P < 0.001$) or as non-significant (ns).

2. The results of the soil C budget (inputs and outputs, Table 1) are very interesting, which are novel compared to other whole-soil-warming experiments in forests. Because the experimental duration is only four years and the effect size is relatively small (0.82% per year of total 0-100 cm SOC stock, line 448-453), the SOC had not been significantly changed by the whole-soil-warming over the ~4 years. Can the authors do a power-analysis and estimate how long (years) would it take to observe a statistically significant change (decrease) in the whole-soil (0-100 cm) SOC stock if the effect size continues?

[Response] Thanks for your constructive suggestions. A simple model (Equation 1) was used to calculate how long (years) whole-soil warming would significantly ($P < 0.05$) change the size of whole-soil SOC stock if the effect size continued.

$$C_{\text{soil}}(t) = C_0 (1-k)^t \quad (1)$$

Where $C_{\text{soil}}(t)$ is the whole-soil SOC stock at any year t , and C_0 is the initial amount of the whole-soil SOC stock. If a 5% loss of SOC stock is assumed to be a significant change, then $C_{\text{soil}}(t)$ is $0.95C_0$. The loss rate (k) is 0.82% per year of total SOC stock. From this simple calculation, we obtain that t is approximately 6.2 years. Therefore, the SOC stock in our study

did not significantly change after 3.3-year whole-soil warming. Perhaps another three years later (August 2024), we may detect a significant decrease in whole-soil SOC stock.

3. For the meta-analysis, can the authors show a prisma diagram (<http://www.prisma-statement.org>)? It would make the results more transparent. A map showing the spatial distribution of the sites for these warming experiments is also helpful. Moreover, it is surprising to me that the warming magnitude is not an important mediator of the warming effect on soil respiration fluxes (Fig. S13, supplementary materials). Can the authors discuss the possible reasons for these results, although a warming study with different levels of warming magnitude is better to answer this question (instead of a meta-analysis which includes results from experiments with different magnitudes, durations, methods, ecosystems, ...).

[Response] Thanks for your reminder and suggestions.

1. We added the Prisma diagram (Supplementary Fig. 17) and site distribution map (Supplementary Fig. 15) of this meta-analysis.
2. In the model-averaged relative importance of the predictor analysis, the cut-off 0.8 is to differentiate the important and non-important predictors (Terrer et al. 2019). But the value of 0.8 is only artificially set to be relative, not absolute. For example, the response of R_a (total data) to warming did not differ between different warming methods (Supplementary Fig. 12), but the warming method is considered an important factor (> 0.8 , Supplementary Fig. 13c). Although the warming magnitude is not an important factor affecting the response of R_s (total data) to warming, we found that there is positive correlation between R_{s_paired} ($P = 0.09$), R_a ($P = 0.05$) and warming magnitude.

However, the meta-analysis was mixed with too many influencing factors, and since the results were derived from different warming experiments (with different methods, durations, and climates), the warming magnitude did not show a significant effect on R_s (total data).

This result is not surprising, as similar results were also found in case studies (Tian 2014; Yu et al. 2019). Maybe the most important reason is that over 55% of the warming duration was less than or equal to 2 years, because we found a significant positive correlation between the response of soil CO_2 flux and warming duration (Supplementary Fig. 14a), so we desperately need to observe a longer time in the future to study the effect of warming magnitude. Finally, the ideal situation is to investigate the effect of warming magnitude in the same experiment with different levels of warming magnitude (which is very costly), especially the whole-soil

warming experiment, which will be more appropriate to study this question. We also supplemented the related discussion (line 229-235) as following:

“However, the warming magnitude did not significantly change the response of total soil CO₂ efflux to warming since the meta-analysis results were derived from different warming experiments (with different methods, durations, and climates), as similar results with previous studies^{30,31}. Maybe the most important reason is that over 55% of the warming duration in this meta-analysis is less than or equal to 2 years. Therefore, the long-term patterns of soil CO₂ emissions and soil C stocks in response to warming need to be further investigated¹².”

Terrer C, Jackson RB, Prentice IC, Keenan TF, Kaiser C, Vicca S, Fisher JB, Reich PB, Stocker BD, Hungate BA, Peñuelas J, McCallum I, Soudzilovskaia NA, Cernusak LA, Talhelm AF, Van Sundert K, Piao SL, Newton PCD, Hovenden MJ, Blumenthal DM, Liu YY, Müller C, Winter K, Field CB, Viechtbauer W, Van Lissa CJ, Hoosbeek MR, Watanabe M, Koike T, Leshyk VO, Polley HW, Franklin O (2019) Nitrogen and phosphorus constrain the CO₂ fertilization of global plant biomass. *Nature Climate Change* 9: 684-689.

Tian LW (2014) The initial impact of simulated warming and simulated grazing to soil respiration and community characteristics of alpine meadow. Qinghai Normal University.

Yu CQ, Wang JW, Shen ZX, Fu G (2019) Effects of experimental warming and increased precipitation on soil respiration in an alpine meadow in the Northern Tibetan Plateau. *Science of the Total Environment* 647: 1490-1497.

Overall, we thank you sincerely for providing these constructive comments and insightful suggestions which improved this work. We hope our revisions and clarifications have satisfactorily addressed your concerns and comments.

Reviewer #3 (Remarks to the Author):

Overall, this is a timely manuscript in which the authors sought to determine how whole-soil (down to 1 meter depth) warming affects carbon cycling in an arctic grassland ecosystem. They also conducted a meta-analysis of warming experiments in grasslands around the globe, but there are many details missing and the presentation of the meta-analysis approach and results are not well integrated into the manuscript.

[Response] Thanks for your constructive comments and insightful suggestions which greatly improved our work. The original purpose of this meta-analysis across global grasslands was to compare the responses of soil CO₂ flux to whole-soil warming (our case study in an alpine grassland) with the general pattern in global responses derived from traditional surface-soil warming. Therefore, the results of our meta-analysis were not described widely in the manuscript. Our detailed responses to your comments are listed below (line numbers refer to the clean version without track changes).

The field experiment is robust and I have no concerns with how it is justified, described, or presented. My biggest concern with this paper is the confusing aspect of the meta-analysis. It is mentioned in the Abstract, Methods, and Supplement, but not the Introduction, Results, or Discussion. It is mentioned as one sentence in the Conclusions section. Where they do describe it, significant details are missing, making it not possible to evaluate the strength of their results. While important, the meta-analysis feels like an “add-on” to the paper, rather than being fully integrated into it. I highly recommend the authors move some of the key aspects of the meta-analysis into the main body of the paper since most readers don’t go to the Supplement or remove it from their paper.

[Response] Thanks for your insightful suggestions. The original purpose of this meta-analysis across global grasslands was to compare the responses of soil CO₂ flux to whole-soil warming (our case study in an alpine grassland) with the general pattern in global responses derived from traditional surface-soil warming. Therefore, the results of our meta-analysis were not described widely in the manuscript. Now, based on your suggestions, we have made the following changes to the meta-analysis section of the manuscript:

1. We added the purpose of this global-scale meta-analysis in the introduction section (line 83-86) as following:

“Moreover, a meta-analysis was made to compare the response of soil CO₂ flux to warming from our novel whole-soil warming experiment in an alpine grassland with the general response pattern in global grasslands derived from traditional surface-soil warming experiments.”

2. We moved the detailed method of our global-scale meta-analysis from the supplementary materials to the main text of the manuscript (see line 491-547).
3. We added a paragraph to the results section to describe the findings of the meta-analysis (line 155-165) as following:

“In the meta-analysis across global grasslands, the effects of experimental warming (none were whole-soil warming) on different types of soil CO₂ efflux (Rs, total; Rh, SOC-derived; Ra, root-derived) were diverse (Fig. 5 and Supplementary Fig. 12). Warming significantly increased Rs by 9% (n = 234, 95% CI: 5%–12%, total data), but the responses of Rs (n = 46, paired data), Rh and Ra (n = 46, for both total and paired data) to warming were not significant (Supplementary Fig. 12). According to the results of Q_B test and model-averaged relative importance of predictors, the method and duration of warming experiments could regulate the response of Rs (total data) to warming (Supplementary Figs. 12-13). Additionally, the effect size of Rs (total data, not the paired data) had a significant positive relationship with the duration of warming (Supplementary Fig. 14a).”

4. In addition to the comparison of the result (our case study with meta-analysis), we also added more discussion on meta-analysis (line 185-186 and 227-234) as following:

“Similarly, experimental warming elevated soil respiration by 9% on average in our global-scale meta-analysis (total data, Fig. 5).”

“In the meta-analysis, we also found that the effect of experimental warming on total soil CO₂ efflux was positively related to warming duration (Supplementary Figs. 13-14, total data). However, the warming magnitude did not significantly change the response of total soil CO₂ efflux to warming since the meta-analysis results were derived from different warming experiments (with different methods, durations, and climates), as similar results

with previous studies^{30,31}. Maybe the most important reason is that over 55% of the warming duration in this meta-analysis is less than or equal to 2 years.”

The following detailed comments are meant to improve the manuscript:

Title and lines 31-32: The authors use the title to summarize “whole-soil warming” and explain that they are the first to do this in an alpine grassland ecosystem, but then they introduce results from a meta-analysis across global grasslands here. I suggest they clarify up front whether the manuscript is also about several arctic grasslands around the globe or just one (their experiment) by either changing the title and text in line 28 or making it clearer in lines 31-32 what studies are included in the meta-analysis that are not a “whole soil warming experiment.”

[Response] Thanks for your suggestions. We modified this sentence into “Moreover, experimental warming only promoted total soil CO₂ efflux by 9% on average in the meta-analysis across global grasslands (non were whole-soil warming).” Based on the suggestions of you and other reviewers, we have revised the title into “Whole-soil warming leads to substantial soil carbon emission in an alpine grassland”.

Related to above comment: The entire paragraph in lines 85-93 only mentions the warming experiment, but not a meta-analysis. I suggest having the Abstract and this paragraph match up in terms of what this paper is about. Since the authors conducted a meta-analysis, that should be included in all aspects of the paper: Title, Introduction, etc.,

[Response] Thanks for your insightful suggestions. Based on your suggestions, we have made the following changes to the meta-analysis section of the manuscript:

1. We added the purpose of this global-scale meta-analysis in the introduction section (line 83-86) as following:

“Moreover, a meta-analysis was made to compare the response of soil CO₂ flux to warming from our novel whole-soil warming experiment in an alpine grassland with the general response pattern in global grasslands derived from traditional surface-soil warming experiments.”

2. We moved the detailed method of our global-scale meta-analysis from the supplementary materials to the main text of the manuscript (line 491-547).

3. We added a paragraph to the results section to describe the findings of the meta-analysis (line 155-165) as following:

“In the meta-analysis across global grasslands, the effects of experimental warming (none were whole-soil warming) on different types of soil CO₂ efflux (Rs, total; Rh, SOC-derived; Ra, root-derived) were diverse (Fig. 5 and Supplementary Fig. 12). Warming significantly increased Rs by 9% (n = 234, 95% CI: 5%–12%, total data), but the responses of Rs (n = 46, paired data), Rh and Ra (n = 46, for both total and paired data) to warming were not significant (Supplementary Fig. 12). According to the results of Q_B test and model-averaged relative importance of predictors, the method and duration of warming experiments could regulate the response of Rs (total data) to warming (Supplementary Figs. 12-13). Additionally, the effect size of Rs (total data, not the paired data) had a significant positive relationship with the duration of warming (Supplementary Fig. 14a).”

4. In addition to the comparison of the result (our case study with meta-analysis), we also added more discussion on meta-analysis (line 185-186 and 227-234) as following:

“Similarly, experimental warming elevated soil respiration by 9% on average in our global-scale meta-analysis (total data, Fig. 5).”

“In the meta-analysis, we also found that the effect of experimental warming on total soil CO₂ efflux was positively related to warming duration (Supplementary Figs. 13-14, total data). However, the warming magnitude did not significantly change the response of total soil CO₂ efflux to warming since the meta-analysis results were derived from different warming experiments (with different methods, durations, and climates), as similar results with previous studies^{30,31}. Maybe the most important reason is that over 55% of the warming duration in this meta-analysis is less than or equal to 2 years.”

The abstract is a bit confusing in regarding to warming-induced effects on SOC. These statements are all made:

-“...whole-soil warming stimulated total and SOC-derived CO₂ efflux by 26% and 37%, respectively”

-“However, the whole-soil SOC pool had not been significantly altered in this study due to the relatively short warming duration (4 years).”

-“We show that whole-soil warming causes a much stronger SOC-climate feedback in the alpine grassland ecosystem than what was reported in previous warming experiments,…”

[Response] We apologize for the misunderstanding.

1. “...whole-soil warming stimulated total and SOC-derived CO₂ efflux by 26% and 37%, respectively” — This is our main finding from the whole-soil warming experiment, and it is easy to understand.

2. “However, the whole-soil SOC pool had not been significantly altered in this study due to the relatively short warming duration (4 years)” — We would like to say that due to the large SOC stock compared to the soil CO₂ emission, four (3.3 exactly) years of whole-soil warming had not significantly changed the size of SOC stock (in 2021).

3. “We show that whole-soil warming causes a much stronger SOC-climate feedback in the alpine grassland ecosystem than what was reported in previous warming experiments, …” — After carefully considering your comments here and later, we decide to change this sentence to “We show that whole-soil warming has a much stronger effect on soil carbon emission in the alpine grassland ecosystem than what was reported in previous warming experiments, most of which only heat surface soils.”

It is understandable that CO₂ will respond more rapidly compared to bulk soil SOC pools, but it is unclear how the authors conclude the last sentence about SOC-climate feedbacks. What evidence are they using to draw this conclusion?

[Response] Thanks for your insightful comments. It is true that soil CO₂ emission is more responsive than soil carbon stock. Therefore, after carefully considering your comments here and later, we decide to drop the “SOC-climate feedback” here and change this sentence to “We show that whole-soil warming has a much stronger effect on soil carbon emission in the alpine grassland ecosystem than what was reported in previous warming experiments, most of which only heat surface soils.”

Moreover, because of the 150-word limitation in the Abstract section required by the journal, we also deleted some sentences and revised the Abstract.

Lines 265-279: There are many important details missing from the description of their meta-analysis methods that are standard when using meta-analysis techniques:

-How many total papers did the Web of Science search bring up?

When they say there were 234 observations, does this mean their search turned up 234 papers, but they used only 72 or that there were 234 observations across the final 72 papers they used? Based on the figures it looks like there were 234 observations (not papers), but this needs to be clearer in the Methods and Results sections.

-How did they treat papers that had multiple time points for soil respiration? I see in Fig. S11 that studies were grouped into three time ranges: < 5, 5-10, and >10 years. Did they average within each paper for these three separate time periods?

-Did they include both lab and field studies? I see they only included experiments, but did these include lab experiments with soils collected from grasslands?

-Were the studies in the meta-analysis constrained to “whole-soil warming experiments” only or no? If not, how are they comparing their “whole-soil warming in an alpine grassland” to “any depth of soil warming to grasslands around the globe”?

-Related to above, did the meta-analysis include warming experiments in alpine grasslands? This would presumably be in a non “whole-soil warming experiment” since theirs is the first, but having a comparison between their experiment and other warming experiments in alpine grasslands would strengthen their paper.

-The field experiment is about “arctic grasslands”, yet the meta-analysis is about grasslands around the globe at all latitudes. It’s unclear why they didn’t match their analyses to the same scale. Since the results of the meta-analysis include grasslands across a range of latitudes, I don’t think it’s fair to compare the 26% loss of SOC-derived CO₂ to 9% found in their grasslands that they mention in the Abstract. I think it’s ok to include this comparison, but then they should also explain what latitudes these values come from, make it clear whether the meta-analysis includes soils that weren’t warmed to such depths, or the comparison it between apples and oranges.

-S12: The 95% confidence intervals across all studies should be shown.

Without this information it is not possible to evaluate the results of their meta-analysis.

[Response] Thanks for your insightful and detailed suggestions.

1. We added the Prisma diagram (Supplementary Fig. 17) to show the process of study selection and a table for all 234 observations from 59 studies (72 papers) included in our meta-analysis (Supplementary Table 3). The database and articles list of this meta-analysis were deposited at <https://github.com/yancyphu/soil-respiration>. We also revised the sentences (line 507-508) as following:

“Finally, 59 studies (from 72 articles) were included in this meta-analysis based on these criteria (Supplementary Fig. 15).”

2. In this meta-analysis, we included multiple years results (not only the latest year result) to obtain more observations. We also revised the sentences (line 517-519) as following:

“Note in this meta-analysis, we included multiple years results (rather than the latest year result) to obtain more observations, particularly for analyzing the effect of warming duration⁵⁶.”

3. We only used the results from field warming experiments (excluding any incubation experiments in the laboratory) in this meta-analysis (line 496-498) as following:

“The key words used for the article search were: (a) field experiment or manipulated experiment (excluding incubation experiments in the laboratory)”

4. In this meta-analysis, all experiments were none whole-soil warming. The original purpose of this meta-analysis across global grasslands was to compare the responses of soil CO₂ flux to warming (this case study, whole-soil warming) with the general pattern in global responses derived from traditional surface-soil warming (that missed the response of deeper soils to warming in grassland ecosystems) (line 492-493) as following:

“To assess the general effect of experimental warming on soil respiration (total, SOC-derived, and root-derived) in global grasslands (none were whole-soil warming)”

5. Of course, these warming experiments in the meta-analysis included warming experiments from alpine grassland ecosystems (but based on non-whole-soil warming). Warming only from alpine grassland ecosystems also significantly increased soil respiration by 13% (n = 94, 95% CI: 7%–20%) in this meta-analysis. In addition, we also compared the results of the whole-soil warming experiment in this study with the results only from alpine grasslands that were based on non-whole-soil warming (line 179-185) as following:

“In a warming experiment conducted in an alpine meadow using infrared heaters to increase surface soil temperature (by 2.5 °C), it was found that warming increased soil respiration by about 14%¹⁸, while a montane meadow experiment using the same technology found that soil respiration decreased by 8% due to 1.6 °C warming¹⁹. In addition, in two recent regional-scale meta-analysis studies, warming promoted soil respiration by about 12% for alpine meadow ecosystems¹⁵, while warming also elevated soil respiration by about 14% for alpine grassland ecosystems²⁰.”

6. We added the site distribution map of this meta-analysis to show study sites from different latitudes (Supplementary Fig. 15). In addition, we also made the frequency distribution histogram of observations among latitude (Fig. SS4). Finally, we also compared the results of the whole-soil warming experiment in this study with the results only from alpine grasslands that were based on non-whole-soil warming.
7. The response ratio of Rs is 0.09 (0.05-0.12, total data) in the meta-analysis (Supplementary Fig. 12). We remade this figure to clearly demonstrate the confidence intervals of the data.

Fig. SS4 The frequency distribution histogram of observations among latitude. See also Supplementary Figure 15 for the site distribution map.

The authors need to include a table of all 234 studies they included in their meta-analysis.

[Response] Thanks for your suggestions. We added a table for all 234 observations included in our meta-analysis (Supplementary Table 3).

Results and Discussion: there is no mention of the results of the meta-analysis. It's unclear why this is missing in these important sections.

[Response] Thanks for your insightful suggestions. Based on your suggestions, we have made the following changes to the meta-analysis section of the manuscript:

1. We added the purpose of this global-scale meta-analysis in the introduction section (line 83-86) as following:

“Moreover, a meta-analysis was made to compare the response of soil CO₂ flux to warming from our novel whole-soil warming experiment in an alpine grassland with the general response pattern in global grasslands derived from traditional surface-soil warming experiments.”

2. We moved the detailed method of our global-scale meta-analysis from the supplementary materials to the main text of the manuscript (see line 491-547).
3. We added a paragraph to the results section to describe the findings of the meta-analysis (line 155-165) as following:

“In the meta-analysis across global grasslands, the effects of experimental warming (none were whole-soil warming) on different types of soil CO₂ efflux (R_s, total; R_h, SOC-derived; R_a, root-derived) were diverse (Fig. 5 and Supplementary Fig. 12). Warming significantly increased R_s by 9% (n = 234, 95% CI: 5%–12%, total data), but the responses of R_s (n = 46, paired data), R_h and R_a (n = 46, for both total and paired data) to warming were not significant (Supplementary Fig. 12). According to the results of Q_B test and model-averaged relative importance of predictors, the method and duration of warming experiments could regulate the response of R_s (total data) to warming (Supplementary Figs. 12-13). Additionally, the effect size of R_s (total data, not the paired data) had a significant positive relationship with the duration of warming (Supplementary Fig. 14a).”

4. In addition to the comparison of the result (our case study with meta-analysis), we also added more discussion on meta-analysis (line 185-186 and 227-234) as following:

“Similarly, experimental warming elevated soil respiration by 9% on average in our global-scale meta-analysis (total data, Fig. 5).”

“In the meta-analysis, we also found that the effect of experimental warming on total soil CO₂ efflux was positively related to warming duration (Supplementary Figs. 13-14, total data). However, the warming magnitude did not significantly change the response of total soil CO₂ efflux to warming since the meta-analysis results were derived from different warming experiments (with different methods, durations, and climates), as similar results with previous studies^{30,31}. Maybe the most important reason is that over 55% of the warming duration in this meta-analysis is less than or equal to 2 years.”

Overall, we thank you sincerely for providing these constructive comments and insightful suggestions which greatly improved this work. We hope our revisions and clarifications have satisfactorily addressed your concerns and comments.

REVIEWER COMMENTS

Reviewer #1 (Remarks to the Author):

The authors of the manuscript "Whole-soil warming leads to substantial soil carbon emission in an alpine grassland" have greatly improved their first submitted manuscript and integrated many of the stated inputs. I see the relevance of the study topic and the data presented. My main remaining concerns are with the statistical methods used and their description in the text which does not allow to either understand completely or reproduce the analysis. See my more detailed comments below:

The authors further explained their upscaling method (lines 290-295). However, in my opinion this is not scientifically robust enough to provide a number as output. The method is oversimplified for such a big statement, the data behind it is not robust enough (4 years from one location) to make a statement about a global scale. I would strongly advise to delete this part in the manuscript.

I agree with your added statement in lines 234-235 and certainly see a great need for long-term experiments. As the name implies, they need a long time to gather information. Perhaps one could think of using existing long-term experiments (e.g. in agricultural systems). Some measure soil temperature and often soil organic content, etc., perhaps this would allow to have long term data, but without the need to specifically run an experiment, as the past years have already shown an effect of climate change, and thus would allow to gather knowledge and improve the design of future experiments.

However, not all effects are just a matter of the duration of the experiment, but also of the "phase". Since the experiment has just started and a drastic change in the environment, such as a whole soil warming, will lead to certain adaptation effects, as I wrote before, one can see in Figure S5 and S7d in the previous version. I think that discussing the effect of such a start would improve the paper and help to interpret the results.

Thank you for your explanation of the statistical method, but I do not understand/feel I can follow your description in lines 475-478. Can you show that this actually reduces autocorrelation? If you chose repeated measures ANOVA, would not a Bonferoni test be the next logical step to show if an where there are actually significant differences? I agree that comparing means over years is a common method, but this is still a comparison of time points, just with an adjusted measurement interval.

I am not convinced by this statistical approach. One assumes that time is an important/driving variable, so it should be corrected for. For example, by first calculating the response to it and then evaluating the differences between treatments, which I think is the standard approach for such dynamic data.

You also mention that you are not achieving the goal of a 4°C increase in your treatment over the entire period. Are you taking the actual temperature differences into account somewhere in your analysis?

Because this will certainly influence the results and may be specific to each individual experimental plot you are using. For example, one could use cumulative temperature (degree days) instead of time over the duration of the experiment to linearize.

In response to your Figure SS1 and SS2, I may not be quite sure what you mean by aboveground biomass and clipped litter. I understood aboveground biomass to be the "untouched" biomass at the time of sampling, while clipped litter was the leftover after the season. However, I assume that this is not the case, because otherwise one would expect a correlation between the two (Figure SS2). Please clarify this in the manuscript to avoid misunderstandings.

Thanks for adding the additional soil type information. However, I think it would be more meaningful to compare the soil depth to other comparable grasslands instead of the global soil depth (lines 337-340).

The use of the SpATS package is not really explained. What did you use as your model, how many knots did you use for the splines, are the residuals normally distributed? What values of the model did you use? I assume you fit the model, used the predict function to get the BLUP (best unbiased prediction) and used these values for further analysis. But this is not mentioned in the text.

You stated that: "Whether our result is one of the outliers (Supplementary Fig. 14) will take longer to verify". I do not quite understand this since you show experiments with fewer years of data in the meta-analysis. So, wouldn't you be able to compare your data (from an entire soil warming experiment) with those you included in your meta-analysis?

Also, the combination/comparison of your specific experiment to the meta analysis is still not perfectly clear to me. For example, you also integrated studies from grasslands which are not alpine grasslands for example, does this have an effect?

Minor:

Not introduced by first appearance: AGB, BGB, BNPP, EOC, ETN, MBC, MBN, BG, NAG, AP, and CUE

Line 174, 176, 195, 251, 267, 269, 289, 292, 295, 334, 339, 340 and 438 different citation style

281 rounding 3.3 years would lead to 3 not 4 in my opinion.

374: define "very similar" (correlation coefficient/RMSE of....?)

551 and 555: data and code links do not work

Reviewer #2 (Remarks to the Author):

I think the authors have addressed most of the comments. I have no further.

Reviewer #3 (Remarks to the Author):

Overall, the authors did a great job revising the manuscript in response to all reviewer comments. However, I have one question and one suggested edit that need to be resolved by the authors.

Line 32 of Abstract: I suggest replacing “(non were whole-soil warming).” with “(none of these experiments were whole-soil warming).”

The authors explain that they did in fact include multiple time points for soil respiration in their meta-analysis, but they do not explain how they treated this statistically. Within each site, the multiple time points are not independent from each other so the authors need to explain how they addressed this.

Reviewer(s)' Comments to Author:

Reviewer #1 (Remarks to the Author):

The authors of the manuscript "Whole-soil warming leads to substantial soil carbon emission in an alpine grassland" have greatly improved their first submitted manuscript and integrated many of the stated inputs. I see the relevance of the study topic and the data presented. My main remaining concerns are with the statistical methods used and their description in the text which does not allow to either understand completely or reproduce the analysis. See my more detailed comments below:

[Response] Thanks for your constructive comments and insightful suggestions which further improved our work. Our detailed responses to your comments are listed below (line numbers refer to the clean version without track changes).

The authors further explained their upscaling method (lines 290-295). However, in my opinion this is not scientifically robust enough to provide a number as output. The method is oversimplified for such a big statement, the data behind it is not robust enough (4 years from one location) to make a statement about a global scale. I would strongly advise to delete this part in the manuscript.

[Response] Thanks for your comments and suggestions. We followed your suggestions and deleted the soil carbon budget and relevant global upscaling calculations (Table 1 and related text).

I agree with your added statement in lines 234-235 and certainly see a great need for long-term experiments. As the name implies, they need a long time to gather information. Perhaps one could think of using existing long-term experiments (e.g. in agricultural systems). Some measure soil temperature and often soil organic carbon content, etc., perhaps this would allow to have long term data, but without the need to specifically run an experiment, as the past years have already shown an effect of climate change, and thus would allow to gather knowledge and improve the design of future experiments.

[Response] Thanks for your insightful comments. We agree with you that it may be possible to detect the effects of climate change on ecosystem processes through long-term observational/monitoring data (e.g. Elmendorf et al. 2015; Liu et al. 2018). In the future, we could also investigate the effects of climate change on soil carbon dynamics through these complementary methods in addition to running manipulative experiments. We have added a sentence to incorporate this idea (line 248-251).

“Alternatively, synchronous monitoring of soil temperature and soil carbon cycling from existing long-term experiments would provide a unique dataset to explore the effects of climate change on soil carbon dynamics.”

Elmendorf SC, et al. (2015) Experiment, monitoring, and gradient methods used to infer climate change effects on plant communities yield consistent patterns. PNAS 112: 448-452.

Liu HY, et al. (2018) Shifting plant species composition in response to climate change stabilizes grassland primary production. PNAS 115: 4051-4056.

However, not all effects are just a matter of the duration of the experiment, but also of the "phase". Since the experiment has just started and a drastic change in the environment, such as a whole soil warming, will lead to certain adaptation effects, as I wrote before, one can see in Figure S5 and S7d in the previous version. I think that discussing the effect of such a start would

improve the paper and help to interpret the results.

[Response] Thanks for your constructive comments. We agree with you that the “phase” of the experiment is important for interpreting the results (e.g. the Harvard Forest soil warming experiment, Melillo et al. 2017). We have added a sentence to incorporate this idea (line 242-246).

“Notably, in addition to the duration of the experiment, the “phase” of the experiment⁵ may also be important for interpreting the effects of climate change on ecosystem processes, since the experiment has just started and a drastic change in the environment, such as a whole-soil warming, will lead to certain adaptation effects.”

Melillo JM, et al. (2017) Long-term pattern and magnitude of soil carbon feedback to the climate system in a warming world. *Science* 358: 101-105.

Thank you for your explanation of the statistical method, but I do not understand/feel I can follow your description in lines 475-478. Can you show that this actually reduces autocorrelation? If you chose repeated measures ANOVA, would not a Bonferoni test be the next logical step to show if and where there are actually significant differences? I agree that comparing means over years is a common method, but this is still a comparison of time points, just with an adjusted measurement interval.

I am not convinced by this statistical approach. One assumes that time is an important/driving variable, so it should be corrected for. For example, by first calculating the response to it and then evaluating the differences between treatments, which I think is the standard approach for such dynamic data.

You also mention that you are not achieving the goal of a 4°C increase in your treatment over the entire period. Are you taking the actual temperature differences into account somewhere in your analysis? Because this will certainly influence the results and may be specific to each individual experimental plot you are using. For example, one could use cumulative temperature (degree days) instead of time over the duration of the experiment to linearize.

[Response] Thanks for your insightful comments and suggestions.

1. The repeated measures ANOVA is commonly used to analyze soil respiration measured over multiple time points, which could indeed reduce autocorrelation (e.g. McCulley et al. 2004; Chang et al. 2016). According to your suggestion, we did the Bonferroni test after repeated measures ANOVA, and also updated the related figures and descriptions in statistical analyses (line 455-456) as following:

“It was performed using the *ezANOVA* function of the “*ez*” R package and then the Bonferroni test was done to assess the effect of warming over time.”

2. As you said, time is an important driving variable. Soil respiration changed over time, but our focus in this study is the effect of warming (rather than the effect of time which is obvious). Therefore, we used statistical methods to detect the effect of warming (such as the commonly used repeated measures ANOVA and the Bonferroni test). Based on your suggestion, we made a new figure (Fig. SS1) to directly show the warming effect on soil CO₂ efflux (% increase) at each measurement time over the four years. Its mean response in each year (growing season) over the four years (2018-2021) was shown in Fig. 4.

Fig. SS1 The effect of whole-soil warming on soil CO₂ efflux (% increase) for growing-season measurements over four years (from June 2018 to September 2021). (a) Rs (soil respiration, total). (b) Rh (heterotrophic respiration, SOC-derived). (c) Ra (autotrophic respiration, root-derived). Error bars are standard errors ($n = 4$). Statistical significance of time is shown by asterisks ($\dagger P < 0.10$, $* P < 0.05$, $** P < 0.01$, $*** P < 0.001$) or as non-significant (ns).

- Our experimental plots were designed in pairs (four pairs, Fig. SS2). Each pair of plots (an ambient plot [CK] and a warming plot [W]) are monitored and controlled independently. The computer program independently controls the temperature increase for each pair of plots (when the temperature increase between CK and W is greater than 4 °C, the power supply is cut off, otherwise it is turned on). And the average temperature increase over the 0-100 cm soil profile (8 different depths) is similar for each pair of plots (pair 1: 3.7 °C, pair 2: 3.7 °C, pair 3: 3.6 °C, pair 4: 3.7 °C). Therefore, we think it is unnecessary to consider the effect of the actual temperature increase of each pair of plots (because it is very similar among the four blocks/paired plots) on soil CO₂ efflux.

Fig. SS2 The plot design in our study. Each pair of plots (an ambient plot [CK] and a warming plot [W]) are monitored and controlled independently.

Chang SX, Shi Z, Thomas BR (2016) Soil respiration and its temperature sensitivity in agricultural and afforested poplar plantation systems in northern Alberta. *Biology and Fertility of Soils* 52: 629-641.

McCulley RL, Archer SR, Boutton TW, Hons FM, Zuberer DA (2004) Soil respiration and nutrient cycling in wooded communities developing in grassland. *Ecology* 85: 2804-2817.

In response to your Figure SS1 and SS2, I may not be quite sure what you mean by aboveground biomass and clipped litter. I understood aboveground biomass to be the "untouched" biomass at the time of sampling, while clipped litter was the leftover after the season. However, I assume that this is not the case, because otherwise one would expect a correlation between the two (Figure SS2). Please clarify this in the manuscript to avoid misunderstandings.

[Response] Thanks for your careful reading and insightful comments. To reduce misunderstandings, we removed "clipped litter" and added "ANPP" (aboveground net primary productivity). ANPP is the total plant dry matter (living biomass + dead detritus) harvested in late August, and AGB (aboveground biomass) is the living plant dry matter harvested in late August. We also revised the sentence of the plant sampling in the Methods section (line 391-393) as following:

"The harvested total plant dry matter (living biomass + dead detritus) was considered to be

ANPP (aboveground net primary productivity), while the harvested living plant dry matter was AGB (aboveground biomass).”

Thanks for adding the additional soil type information. However, I think it would be more meaningful to compare the soil depth to other comparable grasslands instead of the global soil depth (lines 337-340).

[Response] Thanks for your constructive comments. The average soil thickness of alpine grasslands is about 0.7 m (Fang et al. 2010). We replaced "the reference soil depth is also set at 1 m in the Harmonized World Soil Database" with "the average soil thickness of alpine grasslands is about 0.7 m". See line 331-333.

Fang JY, et al. (2010). Ecosystem carbon stocks and their changes in China’s grasslands. *Science China Life Sciences* 53:757-765

The use of the SpATS package is not really explained. What did you use as your model, how many knots did you use for the splines, are the residuals normally distributed? What values of the model did you use? I assume you fit the model, used the predict function to get the BLUP (best unbiased prediction) and used these values for further analysis. But this is not mentioned in the text.

[Response] Thanks for your questions. In the SpATS model, we treated the warming treatment as a fixed effect and the rows of plots (or blocks) as random effects (because in each column, warming and control plots were randomly distributed, and the columns of plots were already considered as potential random effects during the experimental design, Fig. SS2). In addition, the residuals of the predicted values produced by the SpATS model were normally distributed. Please see the code (according to the official introduction of the SpATS package) below for specific parameter settings.

```
“m0 <- SpATS(response = "respiration", spatial = ~ SAP(col, row, nseg = c(1,4)),
             genotype = "geno", ###warming treatment
             random = ~ R,
             data = dat_1, control = list (tolerance = 1e-03))
```

```
pred1.m0 <- predict (m0, newdata = dat_1)
plot(m0)
pred1.m0$residuals<-m0$residuals
pred2.m0 <- predict (m0, which = "geno", predFixed= "marginal")”
```

We added the related sentence in the Methods section (line 459-461) as following:

“We treated the warming treatment as a fixed effect and the rows of plots (or blocks) as random effects (the columns of plots were already considered as potential random effects during the experimental design).”

You stated that: "Whether our result is one of the outliers (Supplementary Fig. 14) will take longer to verify". I do not quite understand this since you show experiments with fewer years of data in the meta-analysis. So, wouldn't you be able to compare your data (from an entire soil warming experiment) with those you included in your meta-analysis?

[Response] Thanks for your comments. Sorry for the confusion about the result comparison in the previous Discussion section. First, we compared our results of the whole-soil warming experiment (from an alpine grassland) with the results of other whole-soil warming experiments

(from various forests, Supplementary Table 1); second, we compared our results of the whole-soil warming experiment (from an alpine grassland) with the meta-analysis results of surface-soil warming experiments (from all grasslands or alpine grasslands only, new Figure 5) globally. Therefore, adding these restrictions would make the comparisons more reasonable. We revised this sentence accordingly.

Also, the combination/comparison of your specific experiment to the meta analysis is still not perfectly clear to me. For example, you also integrated studies from grasslands which are not alpine grasslands for example, does this have an effect?

[Response] Thanks for your comments. To answer your questions, we created a new Fig. 5, which contains the response of soil respiration to warming from all grasslands (a) as well as alpine grasslands only (b). None of these experiments were whole-soil warming. It showed that different types of soil CO₂ efflux (Rs, Rh, and Ra) from alpine grasslands did not respond significantly to warming (Fig. 5b).

The results for all grasslands (Fig. 5a) and alpine grasslands only (Fig. 5b) were generally similar. For example, the mean effect size and confidence interval for Rs_{total} is 6.8% [0.8%, 13.1%] for all grasslands (n = 234) and 8.0% [-3.7%, 21.1%] for alpine grasslands (n = 94).

New Fig. 5 The responses of soil respiration (Rs), heterotrophic respiration (Rh), and autotrophic respiration (Ra) to experimental warming (% increase CO₂ efflux) in all grasslands (a) or alpine grasslands (b) globally (all are surface-soil warming; meta-analysis, total and paired data) and this whole-soil warming in the alpine grassland. Error bars represent 95% confidence intervals (CIs). The vertical dashed line represents the warming effect size = 0. If the 95% CI did not overlap zero, the effect of warming was statistically significant (denoted by *). Total data are from experiments that only measured total soil respiration (but did not separate it into heterotrophic and autotrophic components), while paired data are from experiments that separated total soil respiration into heterotrophic and autotrophic components.

Minor:

Not introduced by first appearance: AGB, BGB, BNPP, EOC, ETN, MBC, MBN, BG, NAG, AP, and CUE

[Response] Thanks for the reminder. We have introduced them at first appearance in this version.

Line 174, 176, 195, 251, 267, 269, 289, 292, 295, 334, 339, 340 and 438 different citation style
[Response] Done. We have used a consistent citation style in this version. In the last version, we used a citation style based on the following paper (also published in *Nature Communications*).

We chose this approach for measuring the temperature response of microbial respiration to eliminate the confounding effects of both differences in substrate availability between soil samples and changes in substrate availability with increasing temperature. Reaction rates in soils are governed by both abiotic (e.g., sorption/desorption) and biotic (i.e., enzyme-regulated activities) processes. Temperature affects both of these processes, but in different ways. Biological reactions are unimodal (i.e., have a temperature optimum), while non-biological reactions typically increase exponentially with increasing temperature¹⁰¹. If C becomes more available at higher temperatures due to increasing sorption/desorption processes (for example), then the measured temperature response would be conflated between these various biotic and abiotic mechanisms. There is also evidence to suggest that substrate availability in soils is a major factor influencing the temperature response of heterotrophic microbial respiration (e.g., ref. 102). By adding an excess of the readily available substrate and subtracting the control at each temperature, we eliminate the issue of more C becoming available at higher temperatures due to increasing abiotic processes and minimise potential discrepancies between soil samples when characterising the microbial temperature response.

information criterion values corrected for a finite sample size (AICc). We found that varying ΔC_p^2 was best or equivalent to keeping it constant for the majority of the soils (Table S8). Thus, we chose to use this version of the model to analyse our temperature response data. We also removed one of the samples from the analysis because the model would not converge during fitting, leaving us with a total of 47 temperature response curves in the final analysis (Fig. S7; Table S9). We estimated the T_{opt} and T_{inf} based on the MMRT curve fits, by identifying the largest predicted value and the largest difference between values generated along the modelled temperature response curves.

Next, we used Moran's I to determine if T_{opt} and T_{inf} varied spatially ($P < 0.0001$). We then fit two spatial simultaneous autoregression (SAR) models using the `spatialreg` package¹⁰⁷ with T_{opt} and T_{inf} as the dependent variables and MET as the independent variable to account for the location effect. SAR models incorporate spatial autocorrelation using a neighbourhood matrix specifying the relationship between the residuals at each site and between neighbours to analyse data with unequal spatial distributions. We also assessed if changes in pH could better explain T_{opt} and T_{inf} than MET using a series of linear regression models for both the geothermal gradient data separately and for the

281 rounding 3.3 years would lead to 3 not 4 in my opinion.

[Response] Thank you for the reminder. Done.

374: define “very similar” (correlation coefficient/RMSE of...?)

[Response] We did a regression of soil CO₂ efflux measured by PS-9000 and that measured by Li-8100 (Fig. SS3). We found that the R² between them was very high (> 0.95), although the value measured by Li-8100 was greater than that by PS-9000. As the main focus of this study is not the absolute values of soil respiration and its components, but their responses to warming, and the fluxes from both ambient and warming plots at each time point were measured by the same equipment, we think changing equipment during the experiment would not significantly affect our main conclusion.

Fig. SS3 The relationship between soil CO₂ efflux measured by PS-9000 (x-axis) and by Li-8100 (y-axis) in July 2019. “Total” means total soil respiration measured from surface collars, while “SOC-derived” means heterotrophic soil respiration measured from deep collars. Note that these two instruments use a similar technology to measure CO₂ concentration and flux. Li-8100 is made in USA and frequently in the field, while PS-9000 is made in China and its core CO₂ sensor is actually bought from the LICOR company.

551 and 555: data and code links do not work

[Response] Sorry about this. In the last version, we gave the editor a temporary link, but it was not accessible to reviewers. In this version, the new link is accessible to everyone.

Overall, we thank you sincerely for providing these constructive comments and insightful suggestions that further improved our work. We hope our revisions and clarifications have satisfactorily addressed your concerns and comments.

Reviewer #2 (Remarks to the Author):

I think the authors have addressed most of the comments. I have no further.

[Response] Thanks for your constructive comments and insightful suggestions which improved our work.

Reviewer #3 (Remarks to the Author):

Overall, the authors did a great job revising the manuscript in response to all reviewer comments. However, I have one question and one suggested edit that need to be resolved by the authors.

[Response] Thanks for your constructive comments and insightful suggestions which further improved our work. Our detailed responses to your comments are listed below (line numbers refer to the clean version without track changes).

Line 32 of Abstract: I suggest replacing “(non were whole-soil warming).” with “(none of these experiments were whole-soil warming).”

[Response] Done.

The authors explain that they did in fact include multiple time points for soil respiration in their meta-analysis, but they do not explain how they treated this statistically. Within each site, the multiple time points are not independent from each other so the authors need to explain how they addressed this.

[Response] Thanks for your questions. Observations from the same study were not independent because they came from different time points, therefore the “study” was treated as a random factor (*rma.mv* function in *metafor*). Also, we updated the related analysis. The main results of the meta-analysis were not changed. We also modified related figures and sentences in the Methods section (line 514-517) as following:

“The *rma.mv* function from the R package “*metafor*” was used to evaluate the weighted effect size and 95% confidence interval (CI) by random-effects models¹⁵. Observations from the same study were not independent because they came from different time points, therefore the “study” was treated as a random factor.”

Overall, we thank you sincerely for providing these constructive comments and insightful suggestions which further improved our work. We hope our revisions and clarifications have satisfactorily addressed your concerns and comments.

REVIEWERS' COMMENTS

Reviewer #1 (Remarks to the Author):

The authors improved their manuscript substantially after the last revisions. Previously, my main concern was with the statistical analysis. I tried to run the analysis using the available code and data. Unfortunately, it is not easy to run it again. The data online is not in the format needed for the provided script. In the spirit of reproducible research, and to provide real value to the reader, I think it is very important that a rigorous and easy to follow analysis is also provided. I would hope, especially for a study published in such a top journal, that the authors would consider providing a working script with at least a subset of the data. I have still not been able to run the full analysis. Also, I still believe that the ezANOVA approach is not entirely correct for this data. However, I think we will remain in disagreement, and some of the effects are visible in the plots with or without any statistical model or p-values applied, so perhaps further discussion will not make too much difference.

The authors used a spatial correction approach, but I did not understand if they only checked the differences between corrected and uncorrected values once. Or if they, as is often done, corrected the raw data, which they then used for the whole statistical analysis. If they did the first, I do not think there would be much of an advantage.

Regarding whether or not to include temperature in the analysis. The authors stated "And the average temperature increase over the 0-100 cm soil profile (8 different depths) is similar for each pair of plots (pair 1: 3.7 °C, pair 2: 3.7 °C, pair 3: 3.6 °C, pair 4: 3.7 °C). Therefore, we consider it unnecessary to consider the effect of the actual temperature increase of each pair of plots (because it is very similar among the four blocks/paired plots) on soil CO₂ efflux". - I do not really understand if these values are the average over time. If so, if I am not mistaken, this may be true, but in terms of the analysis, this is not the measure that could drive the results. If there are larger differences per time point, the repeated measures ANOVA will give different results, because it considers time as a factor and thus calculates an ANOVA per time point.

Figure SS1 helps a lot to get an overview, thanks for adding it. A stronger differentiation between the end of one season and the beginning of a new one would help (i.e. show the gap between September and May more pronounced).

Reviewer(s)' Comments to Author:

Reviewer #1 (Remarks to the Author):

The authors improved their manuscript substantially after the last revisions. Previously, my main concern was with the statistical analysis. I tried to run the analysis using the available code and data. Unfortunately, it is not easy to run it again. The data online is not in the format needed for the provided script. In the spirit of reproducible research, and to provide real value to the reader, I think it is very important that a rigorous and easy to follow analysis is also provided. I would hope, especially for a study published in such a top journal, that the authors would consider providing a working script with at least a subset of the data. I have still not been able to run the full analysis. Also, I still believe that the ezANOVA approach is not entirely correct for this data. However, I think we will remain in disagreement, and some of the effects are visible in the plots with or without any statistical model or p-values applied, so perhaps further discussion will not make too much difference.:

[Response] Thanks for your constructive comments and insightful suggestions which further improved our work.

1. We presented the data in more detail (a figure corresponds to a subset of data and a code script, <https://doi.org/10.6084/m9.figshare.24921495.v2>).
2. The repeated measures ANOVA is a common method for presenting multi-year results of soil respiration and has been used in many similar studies (Wan et al. 2009; Sharkhuu et al. 2016; Soong et al. 2021).

Sharkhuu A, Plante AF, Enkhmandal O, Gonneau C, Casper BB, Boldgiv B, Petraitis PS (2016) Soil and ecosystem respiration responses to grazing, watering and experimental warming chamber treatments across topographical gradients in northern Mongolia. *Geoderma* 269: 91-98.

Soong JL, Castanha C, Pries CEH, Ofiti N, Porras RC, Riley WJ, Schmidt MWI, Torn MS (2021) Five years of whole-soil warming led to loss of subsoil carbon stocks and increased CO₂ efflux. *Science Advances* 7: eabd1343.

Wan SQ, Xia JY, Liu WX, Niu SL (2009) Photosynthetic overcompensation under nocturnal warming enhances grassland carbon sequestration. *Ecology* 90: 2700-2710.

Our detailed responses to your comments are listed below.

The authors used a spatial correction approach, but I did not understand if they only checked the differences between corrected and uncorrected values once. Or if they, as is often done, corrected the raw data, which they then used for the whole statistical analysis. If they did the first, I do not think there would be much of an advantage.

[Response] Thanks for your comments. We only corrected the data of 4-year average soil respiration once by a spatial correction method and then compared the original results (Fig. 3b) with the corrected results (Fig. S9). Because we focused more on the results of 4-year average soil respiration rather than the individual results at different sampling points. However, this approach of spatial correction recommended by reviewer #1 (achieved through the “*SpATS*” R package) is very rarely used (and probably unnecessary) in field studies of soil respiration. In a series of high-level articles on soil respiration (in response to warming) published recently (e.g., Melillo et al. 2017 *Science*; Pries et al. 2017 *Science*; Nottingham et al. 2020 *Nature*; Soong et al. 2021 *Science Advances*), none of them took this approach to correct soil respiration. But, out of respect for reviewer #1, we used this method to correct for 4-year average soil respiration once and found similar results before (Fig. 3b) and after the correction (Fig. S9).

- Melillo JM, Frey SD, DeAngelis KM, Werner WJ, Bernard MJ, Bowles FP, Pold G, Knorr MA, Grandy AS (2017) Long-term pattern and magnitude of soil carbon feedback to the climate system in a warming world. *Science* 358: 101-105.
- Nottingham AT, Meir P, Velasquez E, Turner BL (2020) Soil carbon loss by experimental warming in a tropical forest. *Nature* 584: 234-237.
- Pries CEH, Castanha C, Porras RC, Torn MS (2017) The whole-soil carbon flux in response to warming. *Science* 355: 1420-1422.
- Soong JL, Castanha C, Pries CEH, Ofiti N, Porras RC, Riley WJ, Schmidt MWI, Torn MS (2021) Five years of whole-soil warming led to loss of subsoil carbon stocks and increased CO₂ efflux. *Science Advances* 7: eabd1343.

Regarding whether or not to include temperature in the analysis. The authors stated "And the average temperature increase over the 0-100 cm soil profile (8 different depths) is similar for each pair of plots (pair 1: 3.7 °C, pair 2: 3.7 °C, pair 3: 3.6 °C, pair 4: 3.7 °C). Therefore, we consider it unnecessary to consider the effect of the actual temperature increase of each pair of plots (because it is very similar among the four blocks/paired plots) on soil CO₂ efflux". - I do not really understand if these values are the average over time. If so, if I am not mistaken, this may be true, but in terms of the analysis, this is not the measure that could drive the results. If there are larger differences per time point, the repeated measures ANOVA will give different results, because it considers time as a factor and thus calculates an ANOVA per time point.

[Response] Thanks for your insightful comments. These values (pair 1: 3.7 °C, pair 2: 3.7 °C, pair 3: 3.6 °C, pair 4: 3.7 °C) were average soil temperature over four years. The repeated measures ANOVA is a common method for presenting multi-year results of soil respiration and has been used in many similar studies even though the warming effect on soil temperature may not be exactly the same at different time points (Wan et al. 2009; Sharkhuu et al. 2016; Soong et al. 2021). In this study, we also used the repeated measures ANOVA to test the effects of warming treatment and time (consider the effects of time point) on soil respiration. The results of the repeated measures ANOVA showed that the interaction of warming treatment and measuring time did not significantly affect total soil respiration (Rs) and heterotrophic respiration (Rh). We also corrected the data of 4-year average soil respiration once by the "SpATS" R package to reduce possible heterogeneity (as explained above).

Sharkhuu A, Plante AF, Enkhmandal O, Gonneau C, Casper BB, Boldgiv B, Petraitis PS (2016) Soil and ecosystem respiration responses to grazing, watering and experimental warming chamber treatments across topographical gradients in northern Mongolia. *Geoderma* 269: 91-98.

Soong JL, Castanha C, Pries CEH, Ofiti N, Porras RC, Riley WJ, Schmidt MWI, Torn MS (2021) Five years of whole-soil warming led to loss of subsoil carbon stocks and increased CO₂ efflux. *Science Advances* 7: eabd1343.

Wan SQ, Xia JY, Liu WX, Niu SL (2009) Photosynthetic overcompensation under nocturnal warming enhances grassland carbon sequestration. *Ecology* 90: 2700-2710.

Figure SS1 helps a lot to get an overview, thanks for adding it. A stronger differentiation between the end of one season and the beginning of a new one would help (i.e. show the gap between September and May more pronounced).

[Response] Thanks for your constructive suggestions. According to your suggestion, we showed the gap between September and May (for each year) in Fig. SS1. We also made the same modifications in similar figures (Fig. 2 and Fig. S10).

Fig. SS1 The effect of whole-soil warming on soil CO₂ efflux (% increase) for growing-season measurements over four years (from June 2018 to September 2021). (a) Rs (soil respiration, total). (b) Rh (heterotrophic respiration, SOC-derived). (c) Ra (autotrophic respiration, root-derived). Error bars are standard errors ($n = 4$). Statistical significance of time is shown by asterisks ($\dagger P < 0.10$, $* P < 0.05$, $** P < 0.01$, $*** P < 0.001$) or as non-significant (ns).